# Xbp1 and Brachyury establish an evolutionarily conserved subcircuit of the notochord gene regulatory network

Yushi Wu[†], Arun Devotta[†], Diana S José-Edwards[†], Jamie E Kugler, Lenny J Negrón-Piñeiro, Karina Braslavskaya, Jermyn Addy, Jean-Pierre Saint-Jeannet, Anna Di Gregorio*

Department of Molecular Pathobiology, New York University College of Dentistry, New York, United States

**Abstract** Gene regulatory networks coordinate the formation of organs and structures that compose the evolving body plans of different organisms. We are using a simple chordate model, the *Ciona* embryo, to investigate the essential gene regulatory network that orchestrates morphogenesis of the notochord, a structure necessary for the proper development of all chordate embryos. Although numerous transcription factors expressed in the notochord have been identified in different chordates, several of them remain to be positioned within a regulatory framework. Here, we focus on Xbp1, a transcription factor expressed during notochord formation in *Ciona* and other chordates. Through the identification of Xbp1-downstream notochord genes in *Ciona*, we found evidence of the early co-option of genes involved in the unfolded protein response to the notochord developmental program. We report the regulatory interplay between Xbp1 and Brachyury, and by extending these results to *Xenopus*, we show that Brachyury and Xbp1 form a cross-regulatory subcircuit of the notochord gene regulatory network that has been consolidated during chordate evolution.

**\*For correspondence:**
adg13@nyu.edu

[†]These authors contributed equally to this work

**Competing interest:** The authors declare that no competing interests exist.

## Editor's evaluation

Wu et al. establish the role of *Ciona* X-box binding protein (Xbp1), a basic leucine zipper transcription factor, in notochord morphology, in downstream gene regulation providing novel targets and as an evolutionarily conserved feedback interactor with Bra as shown in *Xenopus*. The manuscript is well written and suggests a conserved regulatory subcircuit between Xbp1 and Bra in *Ciona* and *Xenopus*.

## Introduction

For all chordate embryos, from sea squirts to humans, the notochord represents an essential source of support and patterning signals (*Stemple, 2005*; *Jiang and Smith, 2007*; *Satoh et al., 2012*; *Bagnat et al., 2020*). Studies in organisms representative of all chordate subdivisions have provided evidence that the transcription factors Brachyury and Foxa2 are indispensable for notochord development and constitute an evolutionarily conserved subcircuit of the gene regulatory network (GRN) underlying this process (*Stott et al., 1993*; *Ang and Rossant, 1994*; *Passamaneck et al., 2009*; *Tamplin et al., 2011*; *Di Gregorio, 2020*). Additional transcription factors, either acting downstream of Brachyury and/or Foxa2, or in cooperation with them, control pivotal notochord morphogenetic events, among which the formation of a notochordal sheath consisting of extracellular matrix (ECM) proteins that confer rigidity to the notochord (*Stemple, 2005*; *Bagwell et al., 2020*). One of the most amenable

systems for studies of notochord formation is offered by the ascidian *Ciona*, an invertebrate chordate whose larvae are characterized by a fast-developing and tractable notochord, a compact genome, and unrivaled ease of transgenesis (*Satoh, 2001*; *Di Gregorio and Levine, 2002*; *Stolfi and Christiaen, 2012*). The *Ciona* notochord develops within approximately 1 day after fertilization (*Hotta et al., 2007*); during this time, the high secretory activity of the notochord cells gives rise to the formation of the notochordal sheath, while a fluid-filled lumen forms in the center of the notochord (*Denker, 2012*; *Dong et al., 2009*; *Deng et al., 2013*). The pressure exerted on the rigid notochordal sheath by the lumen provides the tail with a hydrostatic skeleton along which rest the muscle cells flanking the notochord, whose contractions enable the larvae to swim (*Bone, 1992*; *Kier, 2012*). In addition to Brachyury and Foxa2 (Foxa.a in *Ciona*) orthologs, other transcription factors are expressed in the *Ciona* notochord (*Satou et al., 2001*; *Imai et al., 2004*; *Kugler et al., 2008*; *Kugler et al., 2019*; *José-Edwards et al., 2011*; *José-Edwards et al., 2013*; *Reeves et al., 2017*). Among them is the *Ciona* counterpart of X-box binding protein 1 (Xbp1) (*Kugler et al., 2008*), a basic leucine-zipper transcription factor that regulates the unfolded protein response (UPR) (*Mai and Breeden, 1997*; *Yoshida et al., 2001*). The UPR of the endoplasmic reticulum (ER) is an evolutionarily conserved mechanism that allows cells to counteract the stress caused by the presence of improperly folded proteins in the ER (*Yap et al., 2021*). Three ER-stress sensors, Ire1, Perk, and Atf6, are responsible, in metazoans, for the activation of the transcription factors Xbp1, Atf4, and Atf6-alpha, respectively (*Hollien, 2013*; *Mitra and Ryoo, 2019*). In turn, these transcription factors regulate the expression of genes whose products decrease global protein synthesis and enhance the ability of the ER to fold proteins, ultimately restoring proteostasis (*Walter and Ron, 2011*; *Han and Kaufman, 2017*). Physiological processes that challenge the ER, such as an elevated secretory activity, can also activate the UPR; this explains the widespread role of Xbp1 in the development of plasma cells and other cells with sustained secretory activity (*Reimold et al., 2001*; *Iwakoshi et al., 2003*; *Shaffer et al., 2004*;

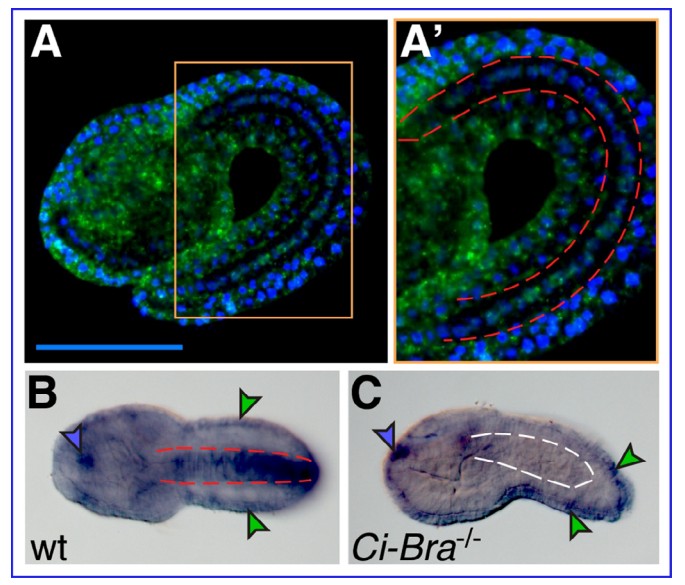

**Figure 1.** *Xbp1* notochord expression is dependent upon Ci-Bra. (**A**) Whole-mount in situ hybridization (WMISH) of a wild-type *Ciona robusta* embryo at the mid-tailbud II stage (*Hotta et al., 2007*) with a fluorescent antisense RNA probe specific for *Cr-Xbp1*. Hybridization signal (green) is visible in both epidermis and notochord cells. (**A'**) Higher-magnification view of the area boxed in light orange in (**A**). The notochord is delineated on both sides by dashed red lines. Nuclei are stained by DAPI (blue; see Materials and methods). Scale bar: 50 µm. (**B**) WMISH of a wild-type *C. robusta* embryo (dorsal view), carried out with a digoxigenin-labeled antisense RNA probe specific for *Cr-Xbp1*. Staining is visible in epidermis (green arrowheads), anterior sensory vesicle (blue arrowhead), and notochord (delineated by dashed red lines). Both RNA probes were synthesized from EST 26p13. (**C**) WMISH of a *Ci-Bra⁻/⁻* mutant *C. robusta* embryo (lateral view) with the same probe described in (**B**). Staining is unaffected in the epidermis (green arrowheads) and sensory vesicle (blue arrowhead), but is lost from the disrupted notochord territory (delineated by dashed white lines).

*Tanegashima et al., 2009*). Loss of *Xbp1* in *Drosophila* is embryonic lethal (*Souid et al., 2007*), and *Xbp1*-knockout mice develop hypoplastic livers and die in utero from anemia (*Reimold et al., 2000*).

Numerous genes controlled by Xbp1 have been identified in plasma cells and pancreatic beta cells (*Acosta-Alvear et al., 2007*); however, the genes directly responsible for the specific role of Xbp1 in notochord morphogenesis, with the notable exception of three chaperone proteins and three proteins that are part of the coat protein I (COPI) complex identified in *Xenopus* (*Tanegashima et al., 2009*), have remained largely unidentified. In addition, the activator(s) responsible for the notochord-specific expression of Xbp1 and the position of this transcription factor within the notochord GRN are still to be elucidated. To bridge these gaps in knowledge, we have analyzed the relationship between *Xbp1* and *Ciona* Brachyury (Ci-Bra; *Corbo et al., 1997*) and studied the effects of alterations in Xbp1 function on notochord development. Through the identification of Xbp1-downstream notochord genes in *Ciona*, we have shed light on a new subcircuit of the notochord GRN, and, through a comparative study, we have found that it is maintained in the vertebrate *Xenopus*.

## Results

### *Cr-Xbp1* notochord expression depends upon Ci-Bra

We had previously identified *Ciona robusta Xbp1* (*Cr-Xbp1*, formerly *Ci-XBPa*; gene model: KH.C4.516) and showed that this gene is predominantly expressed in notochord and epidermis (*Figure 1A and A'*); we had also observed that *Cr-Xbp1* is overexpressed in embryos ectopically expressing Ci-Bra and downregulated in transgenic embryos expressing a repressor form of Ci-Bra (*Kugler et al., 2008*). To directly verify the requirement of Ci-Bra for *Cr-Xbp1* notochord expression, we performed whole-mount in situ hybridization (WMISH) on embryos carrying a null mutation in the *Ci-Bra* coding region (*Chiba et al., 2009*). Compared to stage-matched controls hybridized in parallel (*Figure 1B*), embryos lacking *Ci-Bra* function show normal *Cr-Xbp1* expression in epidermal cells and in a small region of the anterior sensory vesicle, but lack *Cr-Xbp1* expression in the notochord (*Figure 1C*).

### Mutant forms of *Ciona* Xbp1 induce different notochord defects

To investigate the role of Cr-Xbp1 in notochord development, we generated different constructs aimed at interfering with its activity. First, we cloned the region encoding its first 188 amino acid residues downstream of the *Ci-Bra cis*-regulatory region (*Corbo et al., 1997*). This construct expresses in the notochord a truncated Xbp1 protein that retains the leucine-zipper DNA-binding domain (DBD) but lacks the transactivation domain, and is expected to bind its target sequences without activating transcription, according to what has been reported for mouse Xbp1 (*Lee et al., 2003*). The resulting plasmid, *Bra>Xbp1$^{DBD}$::GFP*, was electroporated into *Ciona* zygotes in parallel with a plasmid able to induce the formation of *Xbp1* shRNA and with the neutral notochord marker *Bra>GFP* plasmid (*Corbo et al., 1997*) as a control (*Figure 2A–C*). The resulting transgenic embryos were cultured under the same conditions until they reached the late-tailbud stage, when notochord development was assessed using laser-scanning confocal microscopy. Compared to embryos electroporated with *Bra>GFP* (*Figure 2A*), embryos electroporated with *Bra>Xbp1$^{DBD}$::GFP* had markedly shorter tails, irregularly shaped notochord cells, and displayed aberrant notochord intercalation (*Figure 2B and D*, *Figure 2—figure supplement 1*). In embryos expressing *Xbp1* shRNA, we observed a phenotype milder than the one caused by *Bra>Xbp1$^{DBD}$::GFP*, in a lower percentage of embryos (*Figure 2C and D*).

To determine the effects of the Xbp1 gain-of-function on embryonic development, we cloned its full-length (FL) cDNA downstream of the *Foxa.a* promoter region (*Di Gregorio et al., 2001*). Transgenic embryos carrying the *Foxa.a>Xbp1$^{FL}$* plasmid ectopically express Xbp1 in CNS and endoderm and overexpress it in the notochord. This causes the tail to be bent upward and the notochord cells to be slightly smaller than normal and often arranged into two or more rows throughout the tail (*Figure 2E and F*). Lastly, we generated a presumed hyperactive form of Xbp1 by fusing its DBD to the VP16 transactivation domain (*Sadowski et al., 1988*) (abbreviated as *Bra>Xbp1$^{DBD}$::VP16::GFP*). Compared to stage-matched control embryos (*Figure 2D, G and G'*), embryos carrying the *Bra>Xbp1$^{DBD}$::VP16::GFP* transgene exhibited an evident change in the shape and localization of their notochord cells, which was dependent upon the percentage of transgene incorporation; we also observed an abnormally high number of transgenic cells in the tails of embryos expressing the

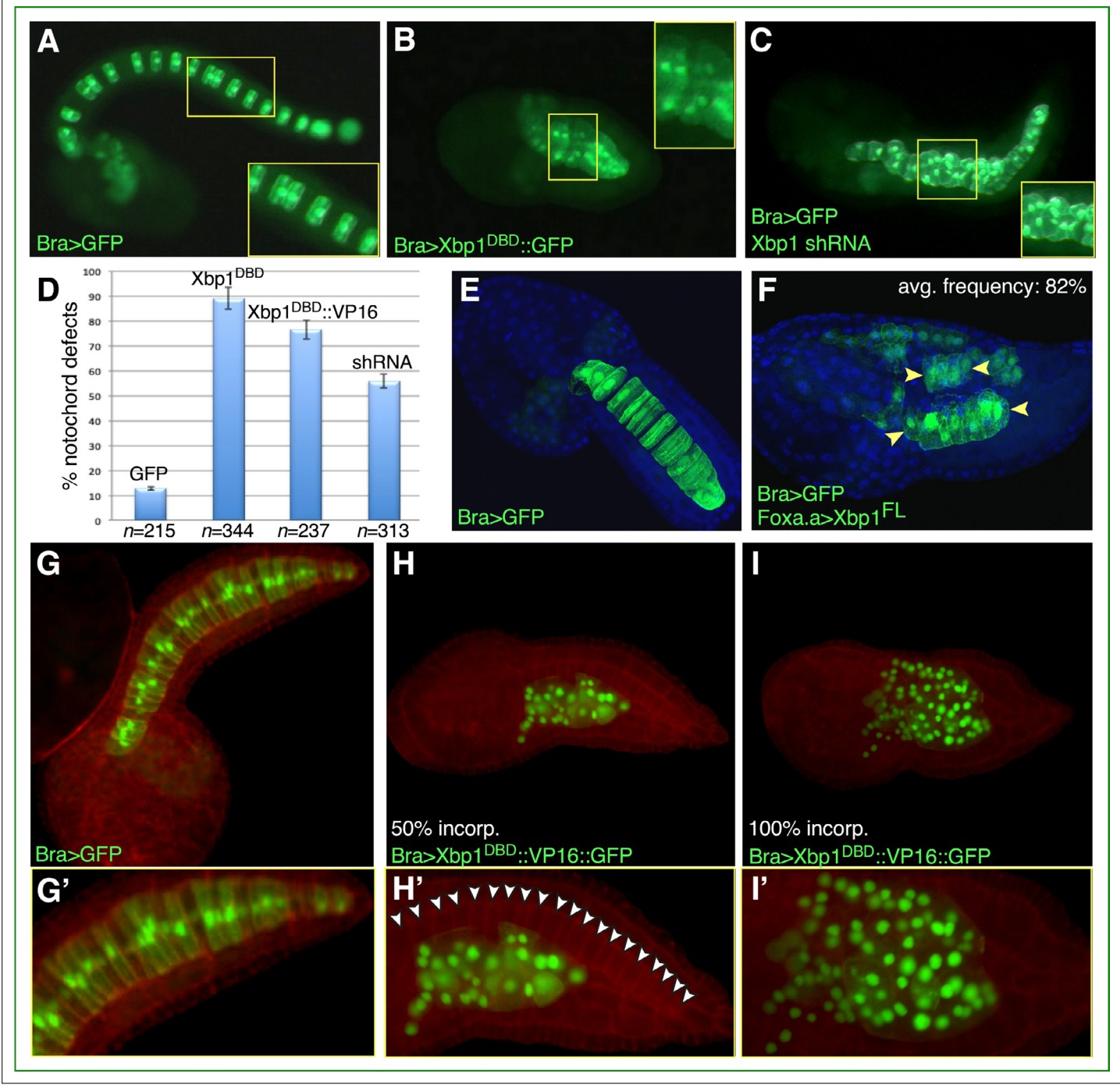

**Figure 2.** Functional analysis of *Ciona* Xbp1. Microphotographs of late-tailbud II (**A–C**), mid-tailbud I (**E, F**), and late-tailbud I (**G–I'**) *C. robusta* transgenic embryos, electroporated at the 1-cell stage with 50 μg of each of the plasmids indicated in the bottom-left corner of each panel. (**A, E, G**) Control embryos electroporated with the notochord marker plasmid *Bra>GFP*, which does not interfere with development (*Corbo et al., 1997*). (**B**) Embryo electroporated with *Bra>Xbp1DBD::GFP*, showing a short tail due to an arrest in notochord development. (**C**) Embryo co-electroporated with *Bra>GFP* and a construct able to express Xbp1 shRNA in the notochord (see Materials and methods), incubated in parallel with the embryos in (**A**) and (**B**), showing a nearly complete tail elongation but defective notochord formation. Insets in (**A–C**) show higher-magnification views of the regions of the notochord boxed by yellow rectangles; all insets display approximately 10 notochord cells, even though in (**A**) mosaic incorporation of the marker plasmid allows clear visualization of only 6 of the 10 selected notochord cells. (**D**) Graph displaying the percentage of embryos showing defective notochord development in *Bra>GFP* control embryos (abbreviated as GFP) and in transgenic embryos carrying *Bra>Xbp1DBD::GFP* (abbreviated as Xbp1DBD), *Bra>Xbp1DBD::VP16::GFP* (abbreviated as Xbp1DBD::VP16), or a *Cr-Xbp1* shRNA construct (abbreviated as shRNA). The total number (n) of fully developed transgenic embryos that were scored per each transgene is reported underneath their respective bars. (**E, F**) Effects of the ectopic/

*Figure 2 continued on next page*

Figure 2 continued

overexpression of Xbp1 in CNS, notochord, and endoderm driven by the *Foxa.a* promoter region. Embryos were stained with DAPI. Avg., average. (**H, I**) Representative embryos carrying the *Bra>Xbp1^DBD^::VP16::GFP* transgene, stained with rhodamine-phalloidin (red). (**H**) Embryo displaying incorporation (incorp.) of the fluorescent transgene in 50% of the notochord cells. (**I**) Representative embryo showing transgene incorporation in the entire notochord lineage (100% incorp.) (**G', H', I'**) Higher-magnification views of the notochord territory of the embryos in (**G**), (**H**), and (**I**), respectively. In (**H'**), white arrowheads indicate the 20 notochord cells (out of 40 total) that have not incorporated the transgene and display a normal morphology.

The online version of this article includes the following figure supplement(s) for figure 2:

**Figure supplement 1.** Additional images of Xbp1^DBD^ and Xbp1^DBD^::VP16 transgenic embryos.

Xbp1^DBD^::VP16::GFP fusion protein (*Figure 2H, I, H', and I'*), which might be caused either by the repositioning of mesenchymal cells from the trunk to the tail or by a loss in the control of cell division in some of the presumptive notochord cells.

## Identification of transcriptional targets of Cr-Xbp1 expressed in the developing notochord of *Ciona*

After analyzing the notochord phenotypes caused by the gain- and loss-of-function experiments described above, we sought to identify the notochord genes that were causing them and that presumably act downstream of Xbp1. To this aim, we collected transgenic embryos from the same clutch, electroporated in parallel with the same amount of DNA (see Materials and methods), expressing either the Xbp1^DBD^::GFP or the Xbp1^DBD^::VP16::GFP fusion proteins, alongside GFP-expressing embryos that were used as controls. RNAs were extracted from each of these three different populations of transgenic embryos, individually labeled, and hybridized to a *C. robusta* microarray. Genes that displayed statistically relevant up- or downregulation in the Xbp1^DBD^::GFP- and in the Xbp1^DBD^::VP16::GFP-expressing embryos compared to the GFP-expressing embryos were selected for further analysis. After these microarray screens were carried out in triplicate, 109 individual putative target genes of Cr-Xbp1 were identified. Expression patterns for 39 of these 109 genes had been previously published (*Brozovic et al., 2018*; *Brozovic et al., 2016*; *Calfon et al., 2002*; *Christiaen et al., 2008*; *Fujiwara et al., 2002*; *Harder et al., 2019*; *Hudson et al., 2011*; *Hudson and Yasuo, 2005*; *Miwata et al., 2006*; *Noiret et al., 2012*; *Ogasawara et al., 2006*; *Parsons et al., 2002*; *Razy-Krajka et al., 2018*; *Shimozono et al., 2010*; *Tetsukawa et al., 2010*), and the expression patterns of 61 of the remaining genes are first described in this study; 9 genes were either not analyzed or provided unclear results (*Supplementary files 1 and 2*). Five genes, *Ci-fibrinogen-like*, *DnaJc7*, *Vps35l*, *Akr1b10*, and *KH.C8.749* (*Supplementary file 1*), were identified in both Xbp1^DBD^::GFP- and Xbp1^DBD^::VP16::GFP-expressing embryos; the limited overlap between the two datasets is likely due to the effect of the VP16 domain on target sequence selection and was observed in previous microarray screens as well (*Butz et al., 2004*; *José-Edwards et al., 2013*). We also noticed that, similarly to Xbp1^DBD^, Xbp1^DBD^::VP16 repressed the expression of several genes, likely through the activation of one or more repressor genes/pathways.

In total, 71 of the 100 expression patterns (71%) include notochord cells, their precursors, and/or broader territories encompassing the notochord (*Supplementary file 1*, *Figure 3*, *Figure 3—figure supplement 1*). The remaining 29 genes are predominantly expressed in mesenchyme (six genes, 6%), epidermis (three genes, 3%), sensory vesicle (five genes, 5%), or any combination of these patterns, often including trunk endoderm, while a few patterns could not be assigned to any specific tissue because their in situ hybridizations produced weak unlocalized staining (*Supplementary file 2*, *Figure 3—figure supplement 2*). The expression of Cr-Xbp1-downstream genes in trunk endoderm and mesenchyme reflects the late expression of Cr-Xbp1 reported in these tissues at the larva stage (*Kusakabe et al., 2002*). According to previous functional studies, morpholino-mediated inactivation of one of the genes downstream of Cr-Xbp1, KH.C12.323, which encodes for an aquaporin channel, causes the disorganization of the body plan (KH.C12.323, cicl027n09; *Hamada et al., 2007*). The Cr-Xbp1 target genes expressed in epidermis include *Pitx*, a well-characterized homeobox gene mainly expressed in the anterior sensory vesicle, epidermis, and oral siphon primordium (*Christiaen et al., 2002*; *Christiaen et al., 2005*), and *Nodal*, the activator responsible for *Pitx* asymmetric expression in the epidermis of the left side of the embryo (*Yoshida and Saiga, 2008*). These findings are consistent with the expression of Cr-Xbp1 in the epidermis of both trunk and tail (*Figure 1A and A'*) and in the anterior sensory vesicle (*Figure 1B and C*). In addition to being expressed during

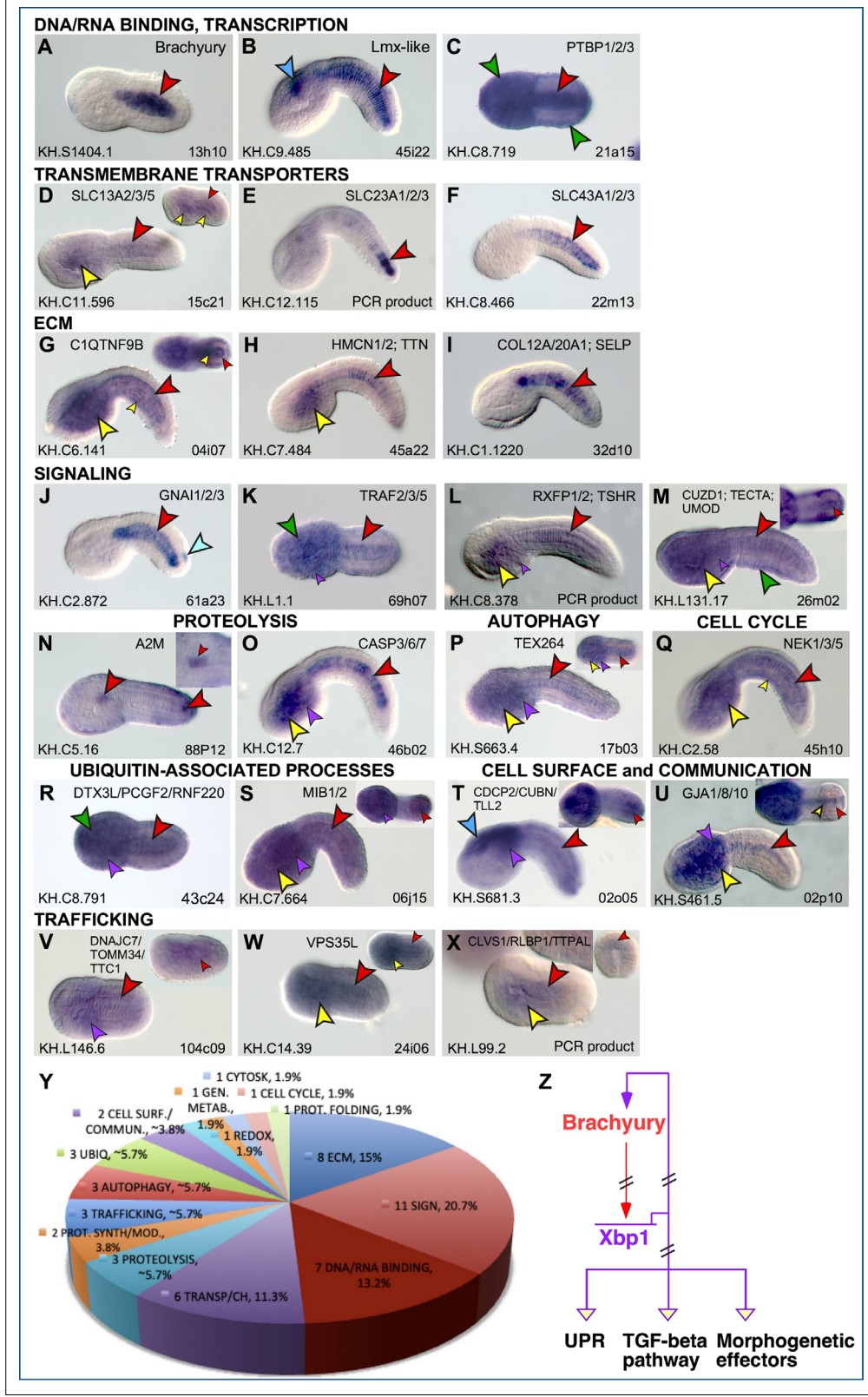

**Figure 3.** Expression patterns of putative Cr-Xbp1 notochord targets. (**A–X**) Whole-mount in situ hybridization (WMISH) of *Ciona* embryos ranging from early gastrula to late tailbud for the genes reported on top of each panel, with digoxigenin-labeled antisense RNA probes synthesized from either the ESTs reported in the lower-right corner of each panel or from gene-specific PCR-amplified products. Gene models are indicated in the

*Figure 3 continued on next page*

*Figure 3 continued*

bottom-left corners. Insets show either embryos at different developmental stages (**D, P, V, W, X**) or optical cross sections of the tails of embryos hybridized with the same probes as those in the main panels (**G, M, S, T, U**). Inset in (**N**) shows a higher magnification of the trunk region of the same embryo, to display staining in the anterior-most notochord cells. Gene ontologies are reported on top of the panel(s) that they refer to (***Supplementary file 1***). Stained territories are denoted by arrowheads, color-coded as follows: red, notochord; blue, CNS; green, epidermis; yellow, endoderm; purple, mesenchyme; orange, muscle; aqua, bipolar tail neuron(s). (**Y**) Pie graph summarizing the gene ontologies of 53 of the 71 potential Cr-Xbp1-downstream genes expressed in the *Ciona* notochord. (**Z**) Schematic representation of the Bra-Xbp1 subcircuit identified by this study and of the processes that it influences in *Ciona*. Filled arrowheads indicate activation of target gene expression, light yellow arrowheads indicate either positive or negative regulation of the genes belonging to each group; slanting parallel lines symbolize that the interactions could be either direct or indirect.

The online version of this article includes the following figure supplement(s) for figure 3:

**Figure supplement 1.** Additional expression patterns of putative Cr-Xbp1 notochord targets.

**Figure supplement 2.** *Ciona*-/ascidian-specific notochord genes regulated by Xbp1.

**Figure supplement 3.** Putative Xbp1 targets expressed in tissues other than the notochord.

---

embryogenesis, Cr-Xbp1 is also detected after metamorphosis, in the newly formed endostyle, a structure homologous to the vertebrate thyroid gland (e.g., ***Sasaki et al., 2003***), in circulating hemocytes, and in part of the digestive tract (***Ogasawara et al., 2002***); accordingly, some of its target genes are expressed in these post-metamorphic structures as well (Tables S2 and S2). In particular, the *serpin* gene that we have detected in the nerve cord (***Figure 3—figure supplement 3***) is expressed after metamorphosis in the endostyle and hemocyte-containing pharyngeal gills, in a pattern that matches the post-metamorphic expression of Cr-Xbp1 (***Ogasawara et al., 2002***).

The 71 presumptive notochord target genes of Cr-Xbp1 include 18 '*Ciona*-/ascidian-specific' genes, which appear to be either specific to *Ciona* (4/18, 22.2%) or to be present in *Ciona* and other ascidians, but currently lack identifiable counterparts in the genomes of organisms from other divisions; nevertheless, five of these ascidian-specific genes (5/18, ~27.8%) contain recognizable protein domains (***Supplementary file 1***, ***Figure 3—figure supplement 2***).

We have grouped the 53 *Ciona* notochord genes with vertebrate counterparts into different categories on the basis of their gene ontologies and the functions of their closest orthologs in other organisms. One of these categories includes seven proteins able to bind nucleic acids (7/53, 13.2%; ***Figure 3A–C***, ***Supplementary file 1***); among them are the notochord transcription factors Ci-Bra and Lmx-like (***Figure 3A and B***; ***Corbo et al., 1997***; ***José-Edwards et al., 2011***), the ubiquitously expressed NRF1/2/6 (***Ishibashi et al., 2003***), and an RNA-binding protein of the PTBP family (***Figure 3C***). Six Cr-Xbp1 targets encode transmembrane transporters and channels (6/53, 11.3%) and include five members of the solute carrier family (SLC; ***Figure 3D–F***) and KCNMB3, a potassium channel (***Satou et al., 2001***; ***Supplementary file 1***). Most of the transmembrane transporters downstream of Cr-Xbp1 belong to the SLC superfamily, which includes $Na^+$-dependent transporters of anionic molecules (***Pizzagalli et al., 2021***). One of these transporters, KH.C11.596, is equally related to human di- and tricarboxylate transporters SLC13A2, SLC13A3, and SLC13A5, another, KH.C12.115, is closer to SLC23A1, SLC23A2, and SLC23A3, all of which transport L-ascorbic acid; KH.C8.466 is equally related to all three members of the small SLC43 subfamily, which includes specialized transporters of neutral amino acids (***Pizzagalli et al., 2021***). Interestingly, the Cr-Xbp1-downstream effectors also include *Ciona* Slc26, an extensively characterized anion transporter necessary for the formation of the central lumen of the notochord during tubulogenesis, the last step of notochord morphogenesis in *Ciona* and other ascidians (***Dong et al., 2009***; ***Deng et al., 2013***).

A significant fraction of Cr-Xbp1 targets encode for ECM proteins (8/53, 15%; ***Figure 3G–I***) and include a hemicentin previously reported as notochord-specific, which we detected also in trunk endoderm, in addition to the notochord (***Figure 3H***, ***Supplementary file 1***), and an extracellular protein equally related to collagen and selectin (***Figure 3I***). Eleven genes encode for signaling molecules (11/53, 20.7%; ***Figure 3J–M***), among which, in particular, transforming growth factor beta (TGF-β) and various components of its signaling pathway, such as Rb1cc1, a mediator of autophagy (***Yao et al., 2021***), olfactomedin2, an ER-localized downstream target of TGF-β (***Shi et al., 2014***), and bone morphogenetic protein BMP4/Lefty1/2, another member of the TGF-β superfamily signaling ligands

(*Supplementary file 1* and references therein). Fibrillin, an ECM coordinator of elastic fibers assembly, is responsible for sequestering TGF-β in the ECM in a latent state, thus regulating its bioavailability (*Dallas et al., 2005*; *Godwin et al., 2019*; *Robertson and Rifkin, 2016*). The signaling molecules controlled by Cr-Xbp1 also include Gnai1/2/3 (G-protein subunit alpha I), a presumed mediator of cell migration expressed in notochord and bipolar tail neurons (*Kim et al., 2020*; *Figure 3J*). Other Gene Ontology (GO) categories include genes presumably involved in various steps of the UPR, such as proteolysis (*Figure 3N and O*), autophagy (*Figure 3P*), cell-cycle regulation (*Figure 3Q*), ubiquitin-associated processes (*Figure 3R and S*), cell communication (*Figure 3T and U*), and trafficking (*Figure 3V–X*). The breakdown of the full complement of 53 Cr-Xbp1 notochord targets into different gene ontologies is provided in *Figure 3Y* and *Supplementary file 1*.

Among the previously published notochord genes targeted by Cr-Xbp1 is *Noto4/PID1*, which was first identified as a downstream target of Ci-Bra (*Takahashi et al., 1999*; *Hotta et al., 2000*), and later on demonstrated to be required for notochord intercalation (*Yamada et al., 2011*). This gene is a direct target of Ci-Bra (*Katikala et al., 2013*) as well as a target of Tbx2/3, which, like Ci-Bra, is a member of the T-box family of transcription factors (*José-Edwards et al., 2013*). The *Noto4/PID1* notochord *cis*-regulatory module (CRM) relies upon a single T-box binding site that is likely targeted by Ci-Bra and/or Tbx2/3 (*Katikala et al., 2013*). Another well-characterized gene that was originally identified as a Ci-Bra target is *fibrinogen-like* (*Takahashi et al., 1999*; *Hotta et al., 2000*), which encodes for a secreted peptide required for the proper positioning of neurons along the developing nerve cord and for axon guidance (*Yamada et al., 2009*). *Noto8*, another direct Ci-Bra target (*Katikala et al., 2013*), encodes for a calmodulin-like protein whose closest counterparts in other organisms act as modulators of motility and ion channel function (*Bennett et al., 2007*; *Inanobe et al., 2015*).

In conclusion, the notochord genes controlled by Cr-Xbp1 can be tentatively grouped into three broad categories: UPR, TGF-β signaling pathway, and morphogenetic effectors sensu stricto; the latter category includes genes directly involved in notochord intercalation and tubulogenesis (*Figure 3Z*). Expression of Cr-Xbp1 is activated in the notochord by Ci-Bra; in turn, Cr-Xbp1 regulates, either directly or indirectly, the expression of its numerous notochord targets and generates a positive feedback loop on the expression of Ci-Bra itself. In addition to sharing some of its target genes with Ci-Bra, Cr-Xbp1 shares part of these notochord genes with Tbx2/3 (*José-Edwards et al., 2013*), and ChIP-chip experiments indicate that the genomic loci of some of the Cr-Xbp1 notochord targets are occupied by Foxa.a in early embryos (*Kubo et al., 2010*; *Supplementary file 1*). No overlap was found between the notochord genes downstream of Cr-Xbp1 and those controlled by another node of the *Ciona* notochord GRN, the ascidian-specific transcription factor Bhlh-tun1 (*Kugler et al., 2019*).

## Mutant forms of Cr-Xbp1 alter the expression of *Ci-Bra* and *fibrillin*

In control *Ciona* embryos carrying the developmentally neutral *Bra>GFP* transgene, the notochord develops normally and *Ci-Bra* mRNA is detected in all of its 40 cells (*Figure 4A–C*). At the mid-tailbud stage, the notochord cells are columnar in shape and exhibit a 'stack of coins' arrangement (*Figure 4C*, *Figure 4—video 1*). In embryos carrying the *Bra>Xbp1^{DBD}::GFP* construct (*Figure 4D–F*), the notochord is mostly composed of irregularly shaped cells, and only the cells that have not received this transgene express *Ci-Bra* (*Figure 4E*). The lack of overlap between the cells carrying the *Bra>Xbp1^{DBD}::GFP* construct (*Figure 4D*) and the cells expressing *Ci-Bra* (*Figure 4E*) indicates that the repressor form of Cr-Xbp1 is able to repress *Ci-Bra* expression (*Figure 4F*, *Figure 4—video 2*). This phenotype is reminiscent of the disruption of notochord development observed in *Ci-Bra* mutants (*Chiba et al., 2009*; *Figure 1C*).

To verify the effect of alterations in the function of Cr-Xbp1 on components of the TGF-β signaling pathway, we analyzed the expression of *fibrillin* in embryos carrying the *Bra>Xbp1^{DBD}::VP16::GFP* transgene (*Figure 4G and H*). As predicted by the results of the microarray screens (*Supplementary file 1*), *fibrillin* is uniformly expressed in the notochord in control embryos (*Figure 4G*), while in *Bra>Xbp1^{DBD}::VP16::GFP* transgenic embryos expression of this gene is limited to the notochord cells that did not incorporate the transgene (*Figure 4H*). This repressive activity of the Xbp1^{DBD}::VP16 fusion is likely indirect and could be due to the activation of a repressor of *fibrillin* expression; alternatively, the Xbp1^{DBD}::VP16 transgene might be solely occupying the Xbp1 binding sites in the regulatory regions of this and other Cr-Xbp1-downstream genes without being able to activate their expression, thus preventing the endogenous Cr-Xbp1 and/or other activators from binding.

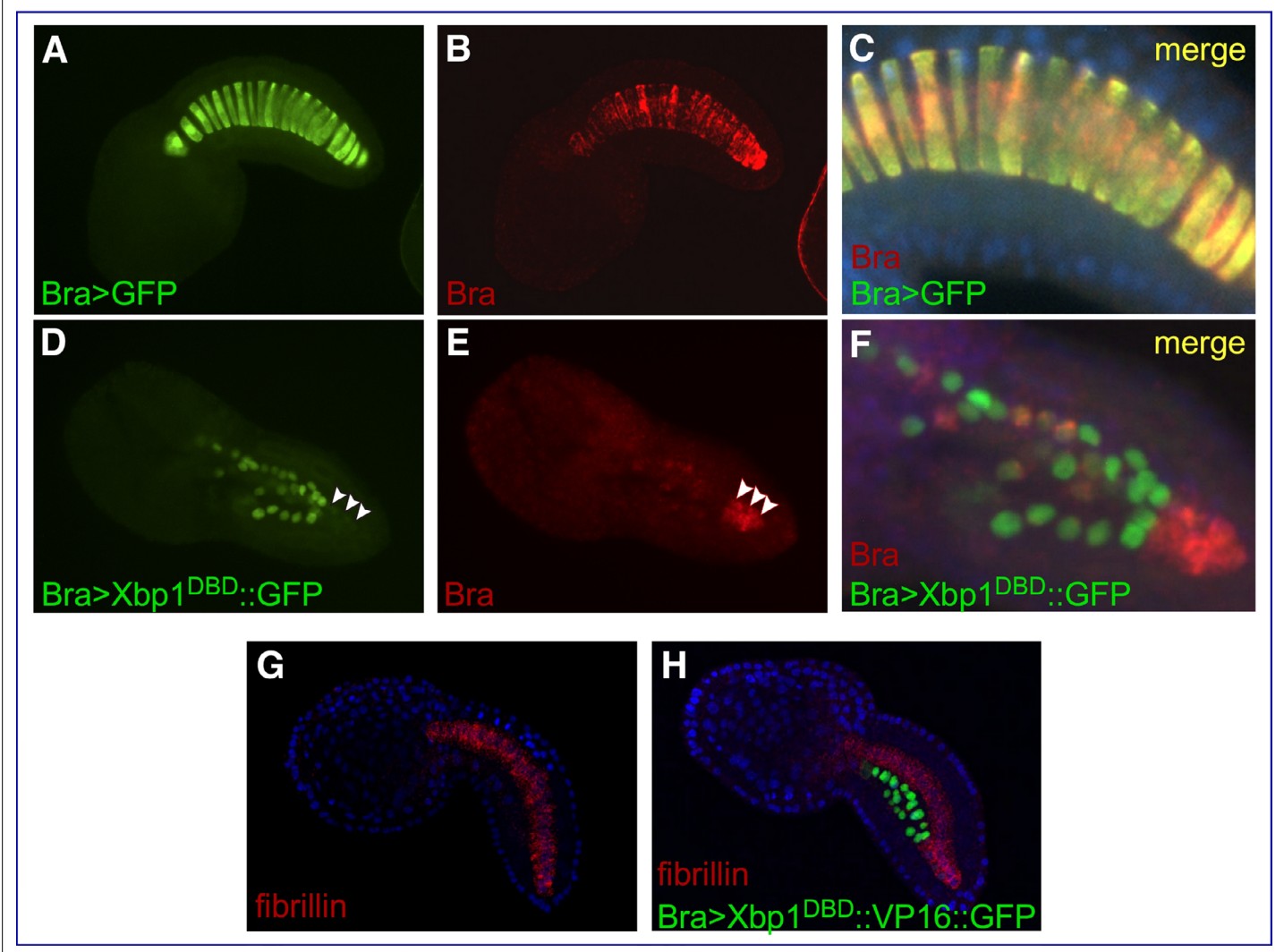

**Figure 4.** Effects of Cr-Xbp1 transgenes on the expression of *Ci-Bra* and *fibrillin*. Mid-tailbud I *C. robusta* embryos electroporated with the transgenes reported in green font and hybridized in situ with fluorescent antisense RNA probes (red font). (**A, B**) *Bra>GFP* transgenic embryo, photographed in the green (**A**) and red (**B**) channels. (**C**) Higher-magnification view of the notochord of the embryo in (**A, B**), obtained after merging the green and red images and the blue channel (DAPI). The *Bra>GFP* plasmid (green) has been incorporated in 20 of the definitive 40 notochord cells (50% incorporation) (**A**). *Ci-Bra* transcripts (red) are detected in all 40 notochord cells (**B, C**). (**D, E**) *Bra>Xbp1DBD::GFP* transgenic embryo displaying mosaic incorporation, photographed in the green (**D**) and red (**E**) channels. White arrowheads indicate a cluster of non-transgenic notochord cells that express Ci-Bra (red). (**F**) Higher-magnification view of the notochord of the embryo in (**D, E**), obtained after merging the green and red images. Expression of *Ci-Bra* is unperturbed in non-transgenic cells (red) and downregulated in transgenic notochord cells (green). (**G, H**) Control wild-type (**G**) and transgenic *Bra>Xbp1DBD::VP16::GFP* (**H**) mid-tailbud I embryos, hybridized in situ with a TRITC-labeled antisense RNA probe for *fibrillin* (gene model: KH.C3.225; EST: 02k18) and counterstained with DAPI. The embryo in (**H**) shows 50% incorporation of the transgene, incomplete notochord intercalation, and downregulation of *fibrillin* in the transgenic notochord cells.

The online version of this article includes the following video for figure 4:

**Figure 4—video 1.** X-projection of confocal images of the embryo in Figure 4C.
https://elifesciences.org/articles/73992/figures#fig4video1

**Figure 4—video 2.** X-projection of confocal images of the embryo in Figure 4F.
https://elifesciences.org/articles/73992/figures#fig4video2

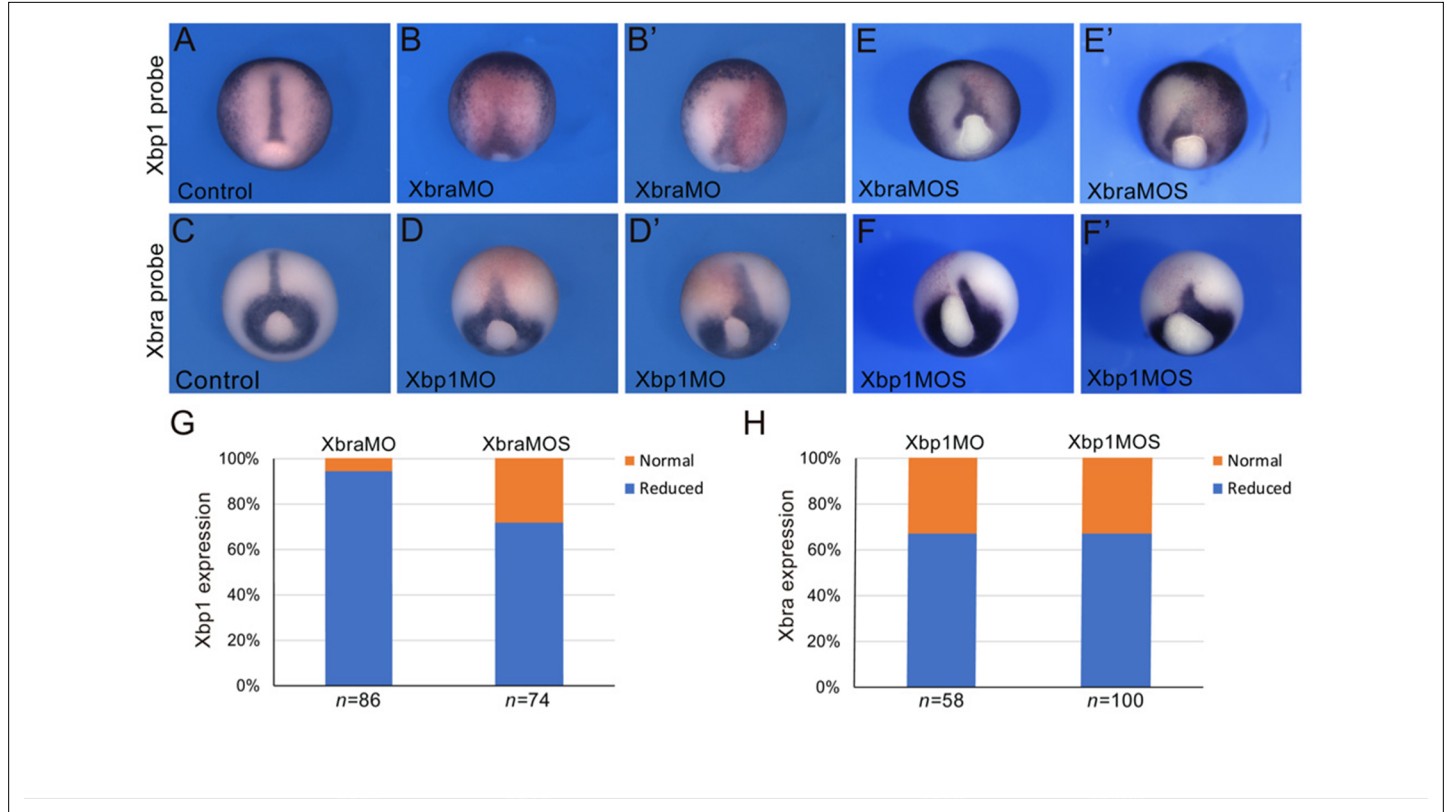

**Figure 5.** Cross-regulation between Brachyury and Xbp1 in the dorsal mesoderm and developing notochord of *Xenopus*. (**A–F'**) *Xenopus laevis* embryos at the late gastrula stage (NF stage 12), control (**A, C**), and morphants (**B, B', D, D', E, E', F, F'**). Injection of the Xbra translation-blocking MO (XbraMO; *Shi et al., 2011*) caused a reduction of *Xbp1* expression in nearly all embryos analyzed (**G**). Xbra splice-blocking MO (XbraMOS) resulted in a similar phenotype, although at a lower frequency, affecting ~70% of the morphant embryos (**G**). Both Xbp1 translation-blocking MO (Xbp1MO; *Yuan et al., 2008*; *Tanegashima et al., 2009*) and splice-blocking MO (XbpMOS) caused a similar reduction of *Xbra* expression in approximately 65% of the morphant embryos (**H**). Embryos are shown as dorsal/vegetal views, anterior to top. The number of embryos analyzed (n) is indicated underneath each bar.

The online version of this article includes the following source data and figure supplement(s) for figure 5:

**Figure supplement 1.** Evaluation of the effects of splice-blocking morpholino oligonucleotides on Xbra and Xbp1 transcripts.

**Figure supplement 1—source data 1.** Original images of the unedited gels presented in *Figure 5—figure supplement 1*.

**Figure supplement 1—source data 2.** Original images of the uncropped gels presented in *Figure 5—figure supplement 1* with the relevant bands labeled.

## The positive cross-regulation between Brachyury and Xbp1 is conserved in *Xenopus*

To extend the results obtained in *Ciona* to higher chordates, we tested whether the positive feedback of Xbp1 on *Brachyury* was conserved in embryos of the amphibian *Xenopus laevis*, a vertebrate in which notochord expression of both these genes had been previously demonstrated (*Smith et al., 1991*; *Zhao et al., 2003*). In *Xenopus*, *Xbra/Tbxt* is expressed throughout the mesoderm during gastrulation and in the prospective notochord at neurula stages (*Smith et al., 1991*). *Xbp1* is first detected in the dorsal lip of the blastopore at the early gastrula stage and persists in the involuting dorsal mesoderm as gastrulation proceeds (*Zhao et al., 2003*). To evaluate the regulation of *Xbp1* by Xbra, we used an antisense morpholino oligonucleotide (MO; *Shi et al., 2011*) to specifically knock down expression of *Xbra* in the dorsal mesoderm. We found that nearly all Xbra morphant embryos showed reduced *Xbp1* expression at the gastrula stage compared to their stage-matched controls (*Figure 5A, B, B' and G*). We also performed the complementary experiment by targeting an Xbp1-specific MO (*Yuan et al., 2008*; *Tanegashima et al., 2009*) to the dorsal mesoderm and analyzing the consequences on *Xbra* expression. Our results show that Xbp1 is implicated in the regulation of *Xbra*

expression in the dorsal mesoderm, as approximately 65% of the Xbp1 morphant embryos displayed reduced *Xbra* expression at gastrula stages (*Figure 5C, D, D' and H*). To confirm the specificity of the knockdown phenotypes, we used a second set of MOs (XbraMOS and Xbp1MOS) that specifically interfere with *Xbra and Xbp1* pre-mRNA splicing, resulting in the production of shorter transcripts due to exon 6 and exon 3 exclusion, respectively (*Figure 5—figure supplement 1*). The phenotype of XbraMOS- (*Figure 5E, E' and G*) and Xbp1MOS-injected embryos (*Figure 5F, F' and H*) was identical to the phenotype generated by the injection of their respective translation-blocking MO. Later in development, unlike stage-matched controls (*Figure 6*), Xbra and Xbp1 morphant embryos displayed axis elongation defects (*Figure 6B and C*) and posterior truncations (*Figure 6B' and C'*). The phenotypes were classified into either 'mild' or 'severe,' and their respective proportions are reported in *Figure 6D*. Overall, these malformations are similar to those observed upon expression of a dominant-interfering Xbra::Engrailed fusion in previous studies (Xbra-En$^R$; *Conlon et al., 1996*). To investigate the evolutionary conservation of the Xbp1-downstream genes identified in *Ciona*, we searched the Xenbase database (Xenbase.org; *Bowes et al., 2010*) and available literature for the expression patterns of *Xenopus* orthologs of *Ciona* notochord genes. We could not find any information on the expression of more than half (27/52; ~52%) of the *Xenopus* putative orthologs of Cr-Xbp1-downstream notochord genes; however, 6 of the remaining genes are reportedly expressed in the notochord, while for the remaining 19 genes expression in this structure has not been reported (*Supplementary file 3*).

Lastly, as a read-out of notochord differentiation, we assessed the expression of the well-characterized notochord marker *Sonic hedgehog* (*Shh*). WMISH of *Xenopus* embryos at NF stage 35/36 with a *Shh* probe indicates that, compared to sibling controls, *Shh* expression is discontinuous in the notochord of Xbra and Xbp1 morphant embryos, and often confined to its posterior regions (*Figure 6—figure supplement 1*). These results suggest that notochord differentiation is impaired by alterations in the Xbra-Xbp1 subcircuit. Altogether, these results indicate that Xbp1 and Xbra regulate each other's expression as part of a regulatory loop controlling not only formation of dorsal mesoderm, but also notochord development and differentiation.

## Discussion

### Cr-Xbp1 acts as a transcriptional intermediary of Ci-Bra during notochord development

In *Ciona* embryos lacking the function of *Ci-Bra* (*Chiba et al., 2009*), expression of Cr-Xbp1 in notochord cells is lost, while its expression in the epidermis and in the anterior region of the sensory vesicle remains unperturbed. These results indicate that Ci-Bra is required for notochord expression of Cr-Xbp1. Embryos carrying transgenes that express either a passive repressor form of Cr-Xbp1 (Xbp1$^{DBD}$) (*Lee et al., 2003*) or shRNA able to interfere with the function of this transcription factor are characterized by a block in the intercalation and differentiation of the notochord cells. These abnormalities resemble those observed in embryos homozygous for a mutation in the *Ci-Bra* coding region (*Chiba et al., 2009*). On the other hand, transgenic embryos in which all notochord cells have incorporated a presumed hyperactive form of Cr-Xbp1 (Xbp1$^{DBD}$::VP16) show numerous round, poorly differentiated transgenic cells dispersed throughout their tails.

To identify the Cr-Xbp1-downstream genes responsible for these phenotypes and to shed light on the molecular mechanisms employed by this transcription factor to regulate notochord development, we performed a transcriptomic analysis of embryos expressing either Xbp1$^{DBD}$::VP16 or Xbp1$^{DBD}$, using embryos expressing GFP as controls. This approach identified 109 genes whose expression is influenced either directly or indirectly by Cr-Xbp1, 71 of which were found expressed in the notochord, either through this study or through the analysis of previously published in situ hybridization data. Eighteen of these genes currently lack clear orthologs in organisms other than *Ciona* and other ascidians. The presence of recognizable protein domains and sequence homology with proteins identified in different organisms allowed the tentative classification of the remaining 53 genes into 15 GO categories. The most ample of these categories include genes whose predicted products either participate in signaling pathways or in the formation of the abundant ECM secreted by the developing notochord cells. 36 of the 71 Cr-Xbp1-downstream notochord genes (50.7%) had been previously reported as targets of Ci-Bra by microarray screens (*Takahashi et al., 1999*; *Hotta et al., 1999*; *Hotta et al.,*

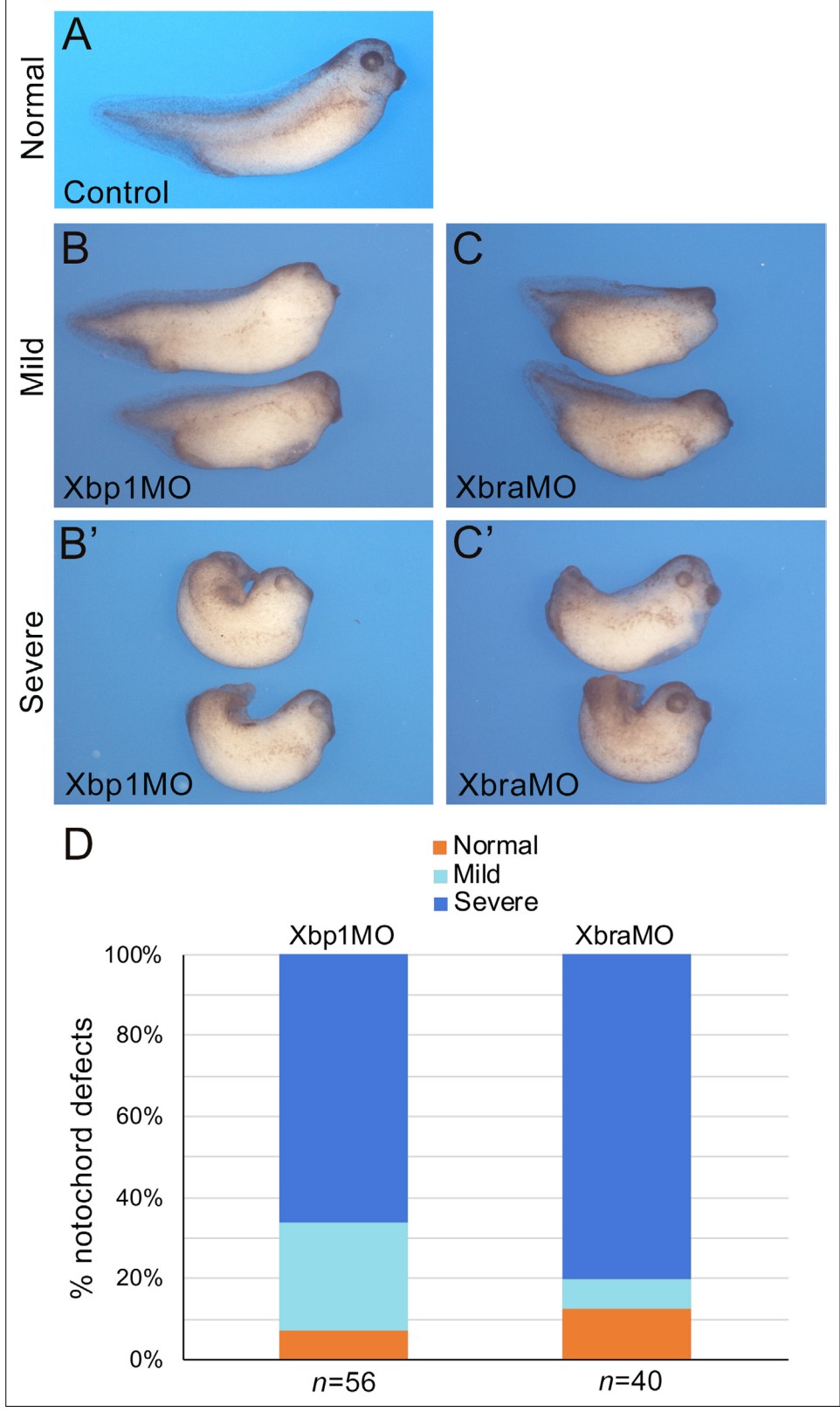

**Figure 6.** Xbra and Xbp1 morpholino-mediated knockdowns result in axis truncation. Xbra (XbraMO) and Xbp1 (Xbp1MO) knockdowns cause comparable anterior-posterior axis elongation defects and posterior truncations. (**A–C′**) *Xenopus* embryos at NF stage 35/36. (**A**) Control embryo. (**B, C**) Morphant embryos injected with Xbp1MO (**B**) and XbraMO (**C**), displaying a mild notochord phenotype. (**B′, C′**) Morphant embryos injected with Xbp1MO

*Figure 6 continued on next page*

*Figure 6 continued*

(**B'**) and XbraMO (**C'**), displaying a severe notochord phenotype. Embryos are shown in lateral views, anterior to the right, dorsal on top. (**D**) Quantification of the notochord defects in Xbp1MO and XbraMO embryos. The number of embryos analyzed (n) is indicated underneath each bar.

The online version of this article includes the following figure supplement(s) for figure 6:

**Figure supplement 1.** Notochord formation in Xbra and Xbp1 morphant embryos monitored through the expression of *Shh*.

---

*2000*), genomic occupancy studies (*Kubo et al., 2010*), and/or RNA-Seq and related transcriptomic experiments (*Reeves et al., 2017*). The extensive overlap between the genes controlled by Cr-Xbp1 and those controlled by Ci-Bra suggests that Cr-Xbp1 might act as a transcriptional intermediary of Ci-Bra during the late stages of notochord development.

## Transcriptional profiling connects Cr-Xbp1 to the UPR, the TGF-β signaling pathway, and notochord morphogenesis

At least 50% of the Cr-Xbp1-downstream genes expressed in the *Ciona* notochord have been reported to participate in the UPR in other model organisms. In addition to participating in the response to ER stress, genes that mediate UPR are also activated under physiological conditions in cells specialized in secretion, such as hepatocytes (*Reimold et al., 2000*), plasma cells (*Iwakoshi et al., 2003*), pancreatic acinar cells (*Lee et al., 2005*), cells of the salivary glands (*Lee et al., 2005*), cells of the hatching gland in fish (*Bennett et al., 2007*), and the notochord cells of zebrafish and *Xenopus* (*Bennett et al., 2007*; *Tanegashima et al., 2009*). We have determined that Cr-Xbp1 controls evolutionarily conserved UPR effectors, such as proteases, chaperones, and other mediators of intracellular trafficking and protein recycling, including Vps35l and Clvs1 (*Singla et al., 2019*; *Katoh et al., 2009*), and the serine/threonine-protein kinase Dapk1, which acts as a UPR sensor and as a mediator of apoptosis and autophagy (*Singh et al., 2016*). Autophagy allows cells to selectively degrade misfolded proteins and defective organelles in response to stress signals (*Kania et al., 2015*); remarkably, this process is also activated in response to hyperosmotic stress in the notochord cells that compose the *nuclei pulposi* of the intervertebral discs (*Jiang et al., 2015*). The present study has identified both inducers and effectors of autophagy expressed in the *Ciona* notochord and regulated by Cr-Xbp1. Ubiquitin ligases, three of which are included among the Cr-Xbp1 targets, are among the main effectors of autophagy and act by tagging unfolded proteins and deteriorating organelles with ubiquitin chains, thus triggering their removal (*Grumati and Dikic, 2018*); accordingly, histone HIST1C/H1.2, another regulator of autophagy (*Wang et al., 2017*), is among the notochord genes controlled by Cr-Xbp1. In addition to autophagy, the UPR can also induce compensatory changes in mitochondrial function (*Senft and Ronai, 2015*), and the presence of mitochondrial metabolic enzymes among the Cr-Xbp1 target genes suggests that this physiological strategy might be present in *Ciona* as well.

Our investigation of the notochord targets of Cr-Xbp1 also connects this transcription factor to the TGF-β signaling pathway. Treatment with TGF-β has been reported to activate expression of Xbp1, and consequently UPR, in mouse and human fibroblasts (*Baek et al., 2012*). Our study indicates that in addition to modulating transcription of *TGF-β* and *BMP4*, Cr-Xbp1 regulates the expression of olfactomedin 2, which in smooth muscle acts downstream of TGF-β to activate a number of tissue-specific markers (*Shi et al., 2014*). We have previously characterized a notochord enhancer region associated with olfactomedin 2 and found that it relies on Fox, homeodomain, and AP1 binding sites for its activity (*José-Edwards et al., 2015*); this indicates that the regulation of this gene by Cr-Xbp1 is likely indirect.

A relevant fraction of the structural genes identified in this study are ECM components. The ECM components include, among others, hemicentins, which are known regulators of cell adhesion (*Xu et al., 2013*), and fibronectin-1, which coordinates ECM assembly and convergent extension (*Halper and Kjaer, 2014*; *Davidson et al., 2006*) and modulates TGF-β signaling together with fibrillin (*Zilberberg et al., 2012*). Of note, the product of another effector of Cr-Xbp1, *fibrinogen-like* (*Hotta et al., 2000*), is secreted in the ECM by the notochord cells and controls the positioning of neurons along the nerve cord through the Notch signaling pathway (*Yamada et al., 2009*). Together with these findings, the notochord phenotypes induced by the transgenes employed in this study suggest that Cr-Xbp1 is

involved in ECM secretion, a crucial step of notochord morphogenesis. The notochord intercalation defect caused by the repressor form of Cr-Xbp1, Xbp1^DBD, is comparable to the phenotype observed in *Ci-Bra* mutants (*Chiba et al., 2009*) and is consistent with the downregulation of *Ci-Bra* that is observed in embryos carrying this transgene. The expression of the Xbp1^DBD::VP16 fusion causes scattering of notochord cells throughout the tail. These considerable defects in notochord formation can be explained by the altered expression of *Noto4/PID1*, which is required for notochord intercalation (*Yamada et al., 2011*), and of several ECM components. In addition to the defective synthesis and secretion of ECM components, which allows the notochord cells to disperse away from the midline, another candidate regulator of this process that is affected by the hyperactive form of Cr-Xbp1 is *Gnai1/2/3*; its corresponding protein has been recently shown to be preferentially localized to the plasma membrane and to the Golgi apparatus, where it might control trafficking of secretory vesicles (*Kim et al., 2020* and references therein). Nek1/3/5, a member of a family of protein kinases involved in the regulation of the cell cycle and mitotic progression (e.g., *Fry et al., 2012*), is upregulated in Xbp1^DBD::VP16 transgenic embryos, which might explain the abnormal number of cells that is occasionally observed in these embryos.

Additionally, Cr-Xbp1 participates in the last steps of notochord morphogenesis, lumen formation and tubulogenesis, by regulating the expression of Slc26, the transmembrane transporter necessary for the expansion of the central lumen of the *Ciona* notochord (*Deng et al., 2013*) and of four additional genes encoding related solute carriers.

## Insights into the role of Xbp1 in notochord development and evolution

Among lower vertebrates, *Xbp1* is expressed in the zebrafish notochord (*Liang et al., 2001*) and studies in medaka fish have shown that the vacuolization of the notochord requires the activity of effectors and transducers of the UPR (*Ishikawa et al., 2017*).

In *Xenopus*, morpholino-mediated knockdown of Xbp1 causes the formation of a smaller than normal notochord (*Tanegashima et al., 2009*). *Xenopus* Xbp1 regulates expression of the chaperone proteins Hsp5A/Bip, DNAJ9B, and HSP90B1 (*Tanegashima et al., 2009*), forms a regulatory loop with BMP-4 in the control of mesoderm and neural differentiation (*Zhao et al., 2003*; *Cao et al., 2006*), and is required for pancreas development (*Yang et al., 2020*). The present study has determined that the regulation of BMP-4 by Xbp1 is present in the *Ciona* notochord as well and has identified, among others, the chaperone protein DnaJc7 as a notochord target of Cr-Xbp1. Most importantly, the results of this analysis have uncovered a regulatory connection between Brachyury and Xbp1 that is maintained in *Xenopus* and is required for the proper development of the notochord in this vertebrate. The information gathered using *Ciona* on the genes influenced by Xbp1 will guide future studies on the notochord genes controlled by *Xenopus* Xbp1.

In mammals, the role of Xbp1 in notochord development remains to be explored. However, a ChIP-on-chip study carried out on chromatin purified from mouse plasma cells, pancreatic beta cells, and skeletal myotubes, both wild-type and subjected to ER stress, has uncovered 545 transcriptional targets of XBP1, most of which form a common core of XBP1-downstream UPR genes that are expressed by all cell types analyzed and are involved in the maintenance of ER homeostasis and control of secretion (*Acosta-Alvear et al., 2007*). At least half of the Cr-Xbp1-downstream genes with traceable vertebrate orthologs identified in this study can be predicted to be participating in the UPR as their gene ontologies correlate with those of XBP1 targets (*Acosta-Alvear et al., 2007*) and cover different facets of this complex process, such as protein folding, proteolysis, trafficking, autophagy, transmembrane transport, and ubiquitination. In mouse embryos, *Xbp1* is expressed in osteoblasts and chondroblasts of several skeletal structures (*Clauss et al., 1993*), and a recent single-embryo, single-cell RNA-Seq study has detected the expression of *Xbp1* in the node/notochord cell population (*Mittnenzweig et al., 2021*). The *Xbp1^{-/-}* mutation is embryonic lethal beginning at day E12.5 (*Reimold et al., 2000*), which leaves open the possibility that Xbp1 mutant mice might have defects in notochord formation as well. Together with the published expression of Xbp1 in the chick notochord (*Bell et al., 2004*; *Darnell et al., 2007*), all these findings and the present study suggest that Xbp1 and UPR genes have been incorporated into notochord formation early during chordate evolution and have been retained in vertebrates as components of the essential notochord developmental program.

Our research on the *Ciona* notochord GRN had previously elucidated the positive feed-forward regulatory input between Brachyury and Tbx2/3 (*José-Edwards et al., 2013*) and the synergistic

control of notochord gene expression by the Bra/Foxa.a subcircuit (*Passamaneck et al., 2009*; *José-Edwards et al., 2015*; *Kugler et al., 2019*; *Di Gregorio, 2020*). Based on these results, on the outcome of morpholino-mediated knockdowns (*Imai et al., 2006*) and on additional evidence, the *Ciona* notochord GRN has been described as being mainly reliant on positive feed-forward interactions (*Reeves et al., 2021*); our results have provided the first report of a positive feedback loop within the *Ciona* notochord GRN and have uncovered a new regulatory subcircuit that links the UPR to notochord development. Furthermore, we have provided evidence that the cross-regulatory interaction between Brachyury and Xbp1 is conserved in the dorsal mesoderm and notochord of *Xenopus*. The regulatory relationship between Brachyury and Xbp1 identified through this research is far-reaching as both transcription factors play crucial roles in a variety of processes that extend far beyond notochord development and include immune response and tumorigenesis.

## Materials and methods

### *Ciona* embryo cultures, electroporation, and imaging

Adult *C. robusta* (formerly *Ciona intestinalis* type A; *Pennati et al., 2015*) were purchased from M-REP (Carlsbad, CA). *C. robusta Brachyury* mutant embryos (originally published as *Ci-Bra* mutants; *Chiba et al., 2009*) were kindly provided by Drs. Shota Chiba and William Smith (U.C. Santa Barbara, CA). Culturing and electroporations were performed as previously described (*Oda-Ishii and Di Gregorio, 2007*). A fraction of the embryos selected for imaging were counterstained with 1U rhodamine-phalloidin (Invitrogen, Carlsbad, CA) in 1X PBS/0.2% Triton X-100, for 3 hr at room temperature. All embryos selected for confocal imaging were mounted using VECTASHIELD with DAPI (Vector Laboratories, USA).

### *Ciona* WMISH

*Ciona* embryos were fixed at stages ranging from gastrula to late tailbud, hybridized, and stained essentially as previously described (*José-Edwards et al., 2011*; *José-Edwards et al., 2013*); whenever necessary, the experiments were repeated at different hybridization temperatures to increase the specificity of the hybridization signal. Fluorescent in situ hybridization and immunostaining were carried out as previously published (*Wagner and Levine, 2012*; *José-Edwards et al., 2013*). Anti-digoxigenin-POD (Roche, IN) and rabbit anti-GFP (Novus Biologicals, CO) antibodies were diluted 1:500 and 1:1000, respectively. In vitro synthesized antisense DIG-labeled RNA probes were visualized using the TSA (Tyramide Signal Amplification) Plus tetramethyl-rhodamine working solution (Perkin-Elmer, MA) for 5–20 min at room temperature, blocked for 1 hr in TNBS (100 mM Tris pH 7.5, 150 mM NaCl, 0.5% Roche blocking reagent, 2% normal goat serum), and incubated at 4°C overnight in the presence of goat anti-rabbit IgG Alexa Fluor 488 secondary antibody (Invitrogen), diluted 1:500. For most genes in this study, antisense RNA probes were synthesized using as templates ESTs from the *Ciona* Gene Collection release 1 (*Satou et al., 2001*) and/or the *Ciona* Unigene cDNA collection (*Gilchrist et al., 2015*; Tables S1 and S2).

### Plasmid construction

The *Bra>Xbp1^{DBD}::GFP* construct was generated by digesting the pFBΔSP6 plasmid (*Oda-Ishii and Di Gregorio, 2007*) with *Xba*I and *Eco*RI to remove the *Ci-Foxa.a* basal promoter and the *LacZ* reporter gene, which were replaced by a linker sequence containing restriction enzyme sites for *Xba*I, *Sac*I, *Kpn*I, *Bsr*GI, and *Eco*RI. This newly created multiple cloning site was digested with *Xba*I/*Sac*I and ligated with the 3.5 kb *C. robusta Brachyury* enhancer/promoter region, lacking the *Ci-Bra* coding sequence (pBraLinker; *Dunn and Di Gregorio, 2009*). Subsequently, a 564bp region encoding for the N-terminal portion of Cr-Xbp1 (aa 1–188), which includes the predicted DBD, was amplified by RT-PCR from RNA extracted from early-tailbud embryos using the QIAGEN RNA miniprep kit (Valencia, CA) as previously described (*Oda-Ishii and Di Gregorio, 2007*) using the primers:

> 5′-Xbp1-SacI: 5′-tgagctcATGAAAATGGCTCCAACCGCTAC-3′ and
> 3′-Xbp1-KpnI: 5′-caggtaccATTCATCAGGAGATAGAATACACTC-3′

(restriction sites are indicated in lowercase), and was cloned downstream of the *Ci-Bra* enhancer/promoter as a SacI-KpnI fragment.

The *Bra>Xbp1^DBD^::VP16::GFP* construct was generated by fusing the DBD of Xbp1 to the VP16 transactivation domain, as previously described (*José-Edwards et al., 2013*; *Sadowski et al., 1988*). The *Cr-Xbp1* shRNA construct was prepared using primers matching nt 9–30 of the *Cr-Xbp1* ORF according to the method reported in *Nishiyama and Fujiwara, 2008*. Primers were annealed, phosphorylated, and ligated into the EcoRI-EcoRV sites of pSP-U6RV (*Nishiyama and Fujiwara, 2008*).

The *Foxa.a>Xbp1^FL^* construct was generated by cloning the complete *Cr-Xbp1* ORF downstream of a 2.5 kb fragment of the *Foxa.a* promoter region (formerly *Fkh/HNF-3beta*; *Di Gregorio et al., 2001*) using the following primers:

> XBP-FL-F NotI: 5′-AAGACAgcggccgcATGAAAATGGCTCCAACCGCTA-3′
> XBP-FL-R BlpI: 5′-ATGTCAgctaagcTTACCACTTTATGAAGAAAATGCAAAAAC-3′

(restriction sites are indicated in lowercase).

## Microarray screens

Approximately 100–300 *C. robusta* embryos were electroporated with 50 µg of either *Bra>Xbp1^DBD^::GFP*, or *Bra>Xbp1^DBD^::VP16::GFP*, or *Bra>GFP* plasmid. Fluorescent transgenic embryos from the same clutch were manually selected in comparable amounts from all three experimental samples using a Zeiss SteReo Discovery V12 epifluorescence microscope at ~6.25 hpf at 21°C, corresponding to initial tailbud I/II according to the standardized developmental table at 18°C (*Hotta et al., 2007*). This time point matches the approximate onset of endogenous *Cr-Xbp1* expression (*Kugler et al., 2008*). Total RNAs were extracted from three biological replicates for each transgenic population using the RNeasy Micro Kit (QIAGEN), and, after being amplified and labeled using the Ambion MessageAmp Premier RNA Amplification Kit (Thermo Fisher Scientific, Waltham, MA), they were hybridized to the *Ciona* Affymetrix GeneChip CINT06a520380F by the Weill Cornell Genomics Resources Core Facility using standard Affymetrix protocols. RNAs extracted from embryos electroporated with the developmentally neutral plasmid *Bra>GFP* were regarded as experimental controls for development after the electroporation procedure and used to determine the extent and significance of up- or downregulation of genes in the Xbp1^DBD^ and XBP1^DBD^::VP16 samples.

Results were RMA summarized from raw data and quantile normalized using GeneSpring GX 11 software by the Weill Cornell Epigenomics Core Facility. Only probe sets with p-values ≤ 0.05 with an absolute fold-change (FC) cut-off of 2.0 were further considered. Changed mRNA levels are expressed as 'up' or 'down' regulated in Tables S1 and S2. The complete dataset has been deposited into the NCBI Gene Expression Omnibus (GEO) under accession number GSE46751.

## *Xenopus Xbp1* probe

*Xenopus Xbp1.L* was amplified by PCR from stage 12 cDNA using the primers:

> F: 5′-ATGGTGGTCGTGGGAGCC-3′
> R: 5′-TTAAAAATGTACATCAAACT-3′

based on the published sequence (*Zhao et al., 2003*). A 1190 bp product was recovered, cloned into the pGEMT vector (Promega, Madison, WI), sequenced, and used to generate in situ hybridization probes. This construct is referred to as pGEMT-Xbp1.

## *Xenopus* embryo injections

The procedures were performed in accordance with the recommendations of the Guide for the Care and Use of Laboratory Animals of the National Institutes of Health, approved by New York University Institutional Animal Care and Use Committee, under animal protocol #150201. *X. laevis* embryos were staged according to *Nieuwkoop and Faber, 1967* (NF) and raised in 0.1× Normal Amphibian Medium (NAM; *Slack and Forman, 1980*). Antisense MOs were purchased from GeneTools (Philomath, OR). We used translation-blocking MOs targeting Xbra/Tbxt and Xbp1. XbraMO (5′-GCGCAGCTCTCGGTCGCACTCATTC-3′) targets both the short (*Xbra.S*) and the long (*Xbra.L*) forms of *Xbra*. For Xbp1 (Xbp1MO), we used a mix (1:1) of two MOs targeting the *Xbp1-S* (5′-GACATCTGGGCCTGCTCCTGCTGCA-3′) and *Xbp1-L* (5′-GCCCAACAAGAGATCAGACTCAGAG-3′). All three translation-blocking MOs have been previously validated (*Shi et al., 2011*; *Yuan et al., 2008*; *Tanegashima et al., 2009*). To further confirm the Xbra and Xbp1 morphant phenotypes, we used a second set of MOs interfering

with *Xbra* and *Xbp1* pre-mRNA splicing. Xbp1MOS (5′-TCTGGAAGAGATCAAACACATGACA-3′) targeting the intron 2/exon 3 junction of both forms of Xbp1, and a mix (1:1) of two MOs:

> 5′-AGTACCTACTGAAGAGAAAGCACAA-3′
> 5′-ACCTACTGAAGGGAAAGCACAAAGA-3′

targeting the intron 5/exon 6 junction of the short and long forms of *Xbra*, respectively (XbraMOS) (*Figure 5—figure supplement 1*). In each experiment, approximately 30 ng of MOs were co-injected with a lineage tracer (*LacZ* mRNA; 0.5 ng) in the equatorial region of both dorsal blastomeres at the four-cell stage (NF stage 3), and the embryos were analyzed by in situ hybridization at NF stage 12 or stage 35/36. Each injection was performed on at least three independent batches of embryos.

### *Xenopus* lineage tracing and WMISH

*Xenopus* embryos at the appropriate developmental stages were fixed in MEMFA (0.1 M 3-N-morpholino-propanesulfonic acid pH 7.4, 2 mM EGTA, 1 mM MgSO$_4$, and 3.7% formaldehyde), stained for Red-Gal (Research Organics; Cleveland, OH) to visualize the lineage tracer (*LacZ* mRNA), and processed for in situ hybridization. Antisense digoxygenin-labeled probes (Genius kit; Roche, IN) were synthesized using as templates cDNAs encoding *Xbra/Tbxt* (*Smith et al., 1991*), *Xbp1* (pGEMT-Xbp1), and *Shh* (*Stolow and Shi, 1995*). WMISH was performed as described (*Harland, 1991*; *Saint-Jeannet, 2017*).

### Morpholino oligonucleotide validation

For MO validation by RT-PCR, total RNAs from control and injected embryos were extracted with RNeasy Micro Kit (QIAGEN) and reverse-transcribed using SuperScript IV VILO Master Mix (Thermo Fisher Scientific) according to the manufacturer's instructions, and used for PCR with Illustra PuReTaq Ready-To-Go PCR beads (GE Healthcare, Chicago, IL). The following primer sets spanning the entire coding sequence were used:

> Xbra_fwd: 5′-ATGAGTGCGACCGAGAGCTG-3′
> Xbra_rev: 5′-TTAGATTGATGGTGGTGCAA-3′
> Xbp1_fwd: 5′-ATGGTGGTCGTGGGAGCC-3′
> Xbp1_rev: 5′-TTAAAAATGTACATCAAACT-3′.

## Acknowledgements

We thank Dr. Elen Gusman, Raymond Li, and Gretchen Neymar Marrero Lozada for their excellent technical help and data analysis. We are indebted to Dr. Izumi Oda-Ishii for sharing her preliminary results on the inactivation of *Ciona* Xbp1. We are grateful to Drs. Shota Chiba and William Smith, UCSB, CA, USA, for the *Ci-Bra* mutant embryos. Research reported in this publication was supported by the Eunice Kennedy Shriver National Institute of Child Health and Human Development of the National Institutes of Health, under award number R03HD098395 and R03HD098395-02S1 to ADG and by a pilot grant to ADG and JPS-J from the New York University Center for Skeletal and Craniofacial Biology, which was established by NIH grant 1P30DE020754. LJN-P was supported in part by NIH training grant T32HD007520. DSJ-E was supported in part by NIH training grant T32GM008539. We thank the NYU Langone Health DART Microscopy Laboratory, which is supported in part by the Cancer Center Support Grant NIH/NCI P30CA016087.

## Additional information

### Funding

| Funder | Grant reference number | Author |
|--------|------------------------|--------|
| National Institutes of Health | R03HD098395 | Yushi Wu<br>Arun Devotta<br>Diana S José-Edwards<br>Jamie E Kugler<br>Lenny J Negrón-Piñeiro<br>Karina Braslavskaya<br>Jermyn Addy<br>Anna Di Gregorio |
| National Institutes of Health | graduate student training grant T32HD007520 | Lenny J Negrón-Piñeiro |
| National Institutes of Health | graduate student training grant T32GM008539 | Diana S José-Edwards |
| National Institutes of Health | Administrative supplement R03HD098395-02S1 | Lenny J Negrón-Piñeiro |
| New York University | Pilot grant | Yushi Wu<br>Arun Devotta<br>Diana S José-Edwards<br>Jamie E Kugler<br>Lenny J Negrón-Piñeiro<br>Karina Braslavskaya<br>Jermyn Addy<br>Jean-Pierre Saint-Jeannet<br>Anna Di Gregorio |
| National Institutes of Health | Center Core Grant for the NYU CSCB 1P30DE020754 | Jean-Pierre Saint-Jeannet<br>Anna Di Gregorio |
| National Institutes of Health | Center Grant for NYU Langone Health DART Microscopy Laboratory P30CA016087 | Yushi Wu<br>Anna Di Gregorio |

The funders had no role in study design, data collection and interpretation, or the decision to submit the work for publication.

### Author contributions

Yushi Wu, Jamie E Kugler, Data curation, Formal analysis, Investigation, Methodology, Validation, Writing – review and editing; Arun Devotta, Data curation, Experiments; Preparation of figures and graphs, Investigation, Validation, Visualization; Diana S José-Edwards, Data curation, Formal analysis, Investigation, Methodology, Validation; Lenny J Negrón-Piñeiro, Data curation, Investigation; Karina Braslavskaya, Data curation, Formal analysis, Investigation, Supervision, Validation; Jermyn Addy, Data curation, Formal analysis, Investigation, Validation; Jean-Pierre Saint-Jeannet, Conceptualization, Data curation, Funding acquisition, Investigation, Methodology, Supervision, Validation, Writing – review and editing; Anna Di Gregorio, Conceptualization, Data curation, Figures preparation, Formal analysis, Funding acquisition, Project administration, Resources, Writing - original draft

### Author ORCIDs

Jean-Pierre Saint-Jeannet (iD) http://orcid.org/0000-0003-3259-2103
Anna Di Gregorio (iD) http://orcid.org/0000-0003-4089-7484

### Ethics

Procedure minimizing discomfort and pain - only applicable to Xenopus. The collection of eggs from females primed with chorionic gonadotropin hormone requires minimum procedures causing virtually no pain or suffering. Surgical dissection of the testes was performed on euthanized males, preventing discomfort. Methods of euthanasia: Male frogs were euthanized after being anesthetized by immersion into a solution of ethyl amino benzoate (tricaine/MS222) through a pithing procedure of the brain and the spinal cord.

Decision letter and Author response
Decision letter https://doi.org/10.7554/eLife.73992.sa1
Author response https://doi.org/10.7554/eLife.73992.sa2

## Additional files

### Supplementary files

- Supplementary file 1. *Ciona* Xbp1 putative target genes expressed in the notochord.
- Supplementary file 2. *Ciona* Xbp1 putative target genes expressed in tissues other than the notochord.
- Supplementary file 3. Expression of *Xenopus* orthologs of *Ciona* Xbp1 notochord target genes.
- Transparent reporting form

### Data availability

The complete dataset has been deposited into the NCBI Gene Expression Omnibus, under accession number GSE46751.

The following dataset was generated:

| Author(s) | Year | Dataset title | Dataset URL | Database and Identifier |
|---|---|---|---|---|
| Di Gregorio A | 2022 | Xbp1 and Brachyury establish an evolutionarily conserved subcircuit of the notochord gene regulatory network | http://www.ncbi. nlm.nih.gov/geo/ query/acc.cgi?acc= GSE46751 | NCBI Gene Expression Omnibus, GSE46751 |

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
