## [Editor Report]

Wu et al. establish the role of *Ciona* X-box binding protein (Xbp1), a basic leucine zipper transcription factor, in notochord morphology, in downstream gene regulation providing novel targets and as an evolutionarily conserved feedback interactor with Bra as shown in *Xenopus*. The manuscript is well written and suggests a conserved regulatory subcircuit between Xbp1 and Bra in *Ciona* and *Xenopus*.

---

## [Decision Letter]

**Decision letter after peer review:**

Thank you for submitting your article "Xbp1 and Brachyury establish an evolutionarily conserved subcircuit of the notochord gene regulatory network" for consideration by *eLife*. Your article has been reviewed by 2 peer reviewers, and the evaluation has been overseen by Marianne Bronner as the Senior and Reviewing Editor. The reviewers have opted to remain anonymous.

The reviewers have discussed their reviews with one another. Both reviewers are enthusiastic about your manuscript but also suggest some additional experiments and changes to the text for clarity. We refer you to their detailed comments, attached below, and ask that you revise the paper accordingly to the best of your ability. We look forward to receiving a revised version of the manuscript.

*Reviewer #2:*

Wu et al. provide evidence that Xbp1 and Bra regulate each other's expression using gene interference in Ciona and *Xenopus*, microarray screening of Xbp1 targets in Ciona upon overexpression of Xbp1-variants and by an individual validation of the endogenous expression in tailbud embryos notably in notochord cells.

The strength of this manuscript is a well written and presented support of a conserved regulatory subcircuit between Xbp1 and Bra in both, Ciona and *Xenopus*. This is convincingly reported in each of the model organism by at least two independent approaches of loss of function. In Ciona, a Bra null mutant affects Xbp1 expression while Xbp1 shRNA or overexpression of truncated Xbp1 variants under the Bra promoter affect notochord cell shape and intercalation while Xbp1 full length under Foxa.a overactivates the Bra promoter. In *Xenopus*, translation blocking and splice blocking MOs reciprocally affect Bra and Xbp1 as well as gastrulation and dorsal blastopore closure. A second Xbp1 target gene is affected in each system, Shh in *Xenopus* and Fibrillin in Ciona.

Furthermore, 109 Xbp1 targets (including Fibrillin) were identified in microarray screening from comparing embryos overexpressing GFP under the Bra promoter or GFP-tagged Xbp1 truncations that contained either the DNA binding domain (DBD) only or its fusion with the VP16 transactivator. Upon individual validation, 80 of these target genes (presented as ISH patterns on tailbud stages) included notochord expression and encompassed several ascidian specific genes. A thorough annotation is provided in two Supplement tables for notochord or non-notochord genes, respectively, and includes knowledge from previous studies notably about regulation by Bra, Foxa.a and Tbx2/3. Overall, three functional groups within Xbp1 targets are proposed that include the UPR (unfolded protein response), TGFß signaling and morphogenetic effectors (such as Fibrillin). This is well discussed in the notochord functional context.

A weakness is that only limited data is presented on endogenous or WT Xbp1 (in Ciona) affecting target gene expression or notochord morphology. Only one type of experiment with full length Xbp1 is presented as a graph of Bra promoter overactivation, but without pictures. In the same line, for the screening, only truncated Xbp1 versions were used, supposing a dominant negative effect of the DBD but of contradictory/unclear effects of VP16. Here, too, no WT Xbp1 was used, neither to analyse notochord morphology.

Concerning the original data sets from microarray screening these are shown as Supplement tables only but not summarized as a Table in the body of the paper. Furthermore the screening logic and expected output are not well explained to the reader causing confusion in particular about the elimination of positive or negative fold change (FC) of expression normally obtained as output of such screens – here changed to absolute values (and deposited as such at NCBI).

In addition, while a potentially valuable interpretation is put forward about different subcircuits regulating notochord formation it is not documented or explained how coregulation by Bra, Tbx2/3 and/or FoxA compares to the proposed independent circuit involving Bhlh-tun1.

Comments for the authors:

To be more convincing and aid the readers in their own judgement of data the following suggestions to eliminate the raised criticisms may help.

1) Lack of Xbp1-WT GOF data:

- Please provide pictures for the Bra promoter activation in Foxa.a driven Xbp1-WT overexpression.

- Please provide a comparative experiment for effects of Xbp1-WT, Xbp1-DBD and Xbp1-VP16.

2) Lack of explanation for partially overlapping effects of DBD or DBD-VP16 (such as on Fibrillin):

- Please more explicitely explain the logic notably seemingly repressive function of Xbp1-VP16 on Fibrillin; is this an indirect activation of a repressor ? please provide a possible interpretation.

3) Lack of explanation for the substractive screening approach and the 'absolute FC' output, also in the source data deposited at NCBI:

- Please more explicitely explain the 'absolute' FC; original (if so) negative or positive FC should probably be kept.

4) Lack of evidence provided for a suggested different Bhlh-tun1 subcircuit:

- Precise examples should be named, listed or a table provided for common and/or different target genes (including corresponding citations).

5) Please add a note in proof on the recent Reeves et al. study from 2021.

*Reviewer #3:*

The paper nicely demonstrates that Xbp1 gene in is involved in notochord development in Ciona. As noted in the manuscript, the role of Xbp1 in notochord development in *Xenopus* is known. In both *Xenopus* and Ciona they show that Xbp1 is acting downstream of Bra in notochord development. It is unclear if Xbp1 is directly or indirectly regulated by Bra.

The impact of this work is to link Bra to Xbp1 in *Xenopus* and Ciona and thereby finding another gene linked to Bra that may be involved in chordate notochord development.

The manuscript focuses on describing lots of genes identified in a subtractive microarray study, there is not enough information to explain what they did. As I am unclear how this was performed it is hard to evaluate the data presented.

In summary, the paper identifies a new gene in Ciona, already known in *Xenopus* and shows that this gene is regulated either directly or indirectly by Bra. It adds to the list of conserved genes involved in notochord development in vertebrate and the non-vertebrate chordate Ciona.

Strengths:

Manipulation of Xbp1 in two different organisms to investigate the role of these gene in notochord development demonstrates that Xbp1 is indeed either directly or indirectly downstream of Bra in notochord development.

This work adds another gene to the network of transcription factors conserved between vertebrate and ciona notochord development.

Weakness:

The data does not address whether or not there is a direct interaction between Bra and Xbp1.

There is no analysis of genes downstream of Xbp1 in *Xenopus* in this study so their statements regarding co-option of genes involved in the unfolded protein response in notochord development is limited to Ciona.

The authors use subtractive microarrays to identify genes downstream of Xbp1. There is hardly any information about this approach and so it is hard to review the resulting data.

There are several places stating data not shown which makes it hard to evaluate.

I found the section on transcriptional targets of Xbp1 hard to follow, simply setting out what the authors are doing would help improve this section so readers can follow. The current description is as follows – "two subtractive microarray screens were carried out in triplicate using the constructs described above".

Please explain what subtractive microarray screens are. What was the purpose of these screens? How were the different constructs shown in the section above used in these microarray screens and in what concentration. How was the data from the two-time points used? What was subtracted from what? How did this approach identify 109 genes? In the methods section it seems there are two-time points at which embryos were collected and three constructs, but I'd like to understand what the analysis was.

There is an assumption that most of the 109 genes are transcriptional targets of Xbp1, but I wonder if the changes in expression could be due to other factors such as loss of notochord cell identity or response to electroporation of DNA into the embryos. It is hard to evaluate the claims without understanding what was done.

The authors suggest through identification of Xbp1-downstream notochord genes they find evidence of the early co-option of genes involved in the unfolded protein response to the notochord development program. In the text it lists all the genes that are differentially regulated in their analysis -with relation to the UPR – "Other gene ontology categories include genes presumably involved in the UPR, such as protein folding…" This is shown in figure 3. What % of the 109 genes are UPR genes. Could differential expression of UPR genes be a result of the way the experiment was done and the use of over expression constructs? Are these genes up or down regulated by the Bra>Xbp1dbd::GFP or Bra>Xbp1::VP16::GFP. I am aware that the control has GFP, but what is the size of the coding regions of the other constructs, how much was electroporated, I could not find the ug of constructs used for these experiments. I would like to know more about how the experiment was done to rule out the possibility that the UPR genes were differentially expressed as a result of electroporation of these constructs.

The text should clearly point out that their observations regarding the co-option of genes involved in the unfolded protein response in notochord development relates to Ciona only. In the manuscript the only data I see on genes downstream of Xbp1 studied in *Xenopus* is Shh. In the discussion they mention Bmp being known to be perturbed by Xbp1 knock down in *Xenopus* and that similar results were seen in Ciona.

It would be helpful to know if the genes identified as downstream of Xbp1 in the Ciona are also expressed in the *Xenopus* notochord. Could they have a figure showing what fraction of the 109 genes identified are also found in the *Xenopus* notochord?

Xbp1 knock out mice have been studied. In the publications about these mice are there any notochord defects mentioned? If in the knockout mice there are notochord defects this should be stated. If there are notochord defects this provides evidence in another vertebrate of the role of Xbp1 in notochord development.

The authors state that Xbp1 in the mammalian notochord has not been studied. There is ample single cell RNA-seq datasets from embryos in mammalian systems in which the authors could look for notochord expression of Xbp1. It would help the manuscript if this data was utilized to identify if expression of this gene is conserved in other vertebrates.

Since Xbp1 has already been studied in the *Xenopus* notochord, it would be helpful to state what has been learned from this study that was not appreciated in the previous analysis of Xbp1 in *Xenopus* to clearly show the knowledge added by this study.

There are several statements regarding data followed by data not shown. The data should be shown.

---

## [Author Response]

Reviewer #2:[…] (1) Lack of Xbp1-WT GOF data:- Please provide pictures for the Bra promoter activation in Foxa.a driven Xbp1-WT overexpression.

Over the past months we have identified putative Xbp1 binding sites in the 3.5-kb *Ci-Bra* promoter, and we have mutagenized a few of them in an effort to determine whether there is a direct interaction between Xbp1 and the Ci-Bra promoter. Since we have not obtained conclusive evidence of such interaction, and there are still numerous putative binding sites that would have to be mutagenized in order to have a definitive answer, we have removed this result from the revised manuscript. However, we did insert a microphotograph of an embryo displaying the phenotype induced by the Foxa.a-driven full-length Xbp1 overexpression (revised Figure 2E,F; see below).

- Please provide a comparative experiment for effects of Xbp1-WT, Xbp1-DBD and Xbp1-VP16.

In revised Figure 2, we have added a microphotograph of an embryo expressing Xbp1 shRNA (panel C). We have also added two panels (E,F) to show the results of the ectopic expression of the full-length (FL) wild-type Xbp1 protein in CNS and endoderm, through the Foxa.a promoter region; this construct (Foxa.a>Xbp1^FL^) also induces an overexpression of Xbp1 in the notochord. In sum, the revised Figure 2 now displays a comparison of the effects of the overexpression of Xbp1-WT, Xbp1-DBD and Xbp1-VP16 in the developing notochord.

2) Lack of explanation for partially overlapping effects of DBD or DBD-VP16 (such as on Fibrillin):- Please more explicitely explain the logic notably seemingly repressive function of Xbp1-VP16 on Fibrillin; is this an indirect activation of a repressor ? please provide a possible interpretation.

We have added the interpretation of the repressive function of Xbp1::VP16 on pages 8 and 13 of the revised manuscript.

3) Lack of explanation for the substractive screening approach and the 'absolute FC' output, also in the source data deposited at NCBI:- Please more explicitely explain the 'absolute' FC; original (if so) negative or positive FC should probably be kept.

We have added a detailed explanation of the goals and details of the microarray screens to the text. As per the fold-change, we wish to clarify that the results were provided in the ‘absolute FC’ format after being analyzed and statistically validated, therefore there is no ‘original’ positive or negative FC. The data that we report in the manuscript have been kept consistent with the source data that we have deposited in the NCBI GEO database. For these reasons, we have maintained this format in the manuscript as well.

4) Lack of evidence provided for a suggested different Bhlh-tun1 subcircuit:- Precise examples should be named, listed or a table provided for common and/or different target genes (including corresponding citations).

We have clarified this concept by rephrasing the sentence as follows: “No overlap was found between the notochord genes downstream of Cr-Xbp1 and those controlled by another node of the *Ciona* notochord GRN, the ascidian-specific transcription factor Bhlh-tun1 (Kugler et al., 2019).”

All the information currently available on Bhlh1 and its target genes in the notochord and corresponding citations is included in our published work (Kugler et al., 2019).

5) Please add a note in proof on the recent Reeves et al. study from 2021.

We have added a citation of this paper and how it might relate to our findings in the Discussion section, on page 22 of the revised text.

Reviewer #3:[…] I found the section on transcriptional targets of Xbp1 hard to follow, simply setting out what the authors are doing would help improve this section so readers can follow. The current description is as follows – "two subtractive microarray screens were carried out in triplicate using the constructs described above".Please explain what subtractive microarray screens are. What was the purpose of these screens? How were the different constructs shown in the section above used in these microarray screens and in what concentration. What was subtracted from what? How did this approach identify 109 genes?

We have removed “subtractive” from the text and we have clarified the purpose and the methodology of these screens, and how this approach identified 109 genes (page 8). We have added the concentration of the constructs (50 micrograms) and further explanation in the Methods section (page 23-24) and in the legend to figure 1.

How was the data from the two-time points used? What was subtracted from what? How did this approach identify 109 genes? In the methods section it seems there are two-time points at which embryos were collected and three constructs, but I'd like to understand what the analysis was.

We are thankful to this Reviewer for spotting our mistake in the Methods section. Differently from our previously published work on Tbx2/3, the Xbp1 microarray screen did not involve two experimental time points but only one. We have corrected this mistake in the revised manuscript.

There is an assumption that most of the 109 genes are transcriptional targets of Xbp1, but I wonder if the changes in expression could be due to other factors such as loss of notochord cell identity or response to electroporation of DNA into the embryos. It is hard to evaluate the claims without understanding what was done.

We have added a detailed explanation of the goals and details of the microarray screens to the revised text and we explain below the reasons why we believe that the results of this screen are specific (please see below).

The authors suggest through identification of Xbp1-downstream notochord genes they find evidence of the early co-option of genes involved in the unfolded protein response to the notochord development program. In the text it lists all the genes that are differentially regulated in their analysis -with relation to the UPR – "Other gene ontology categories include genes presumably involved in the UPR, such as protein folding…" This is shown in figure 3. What % of the 109 genes are UPR genes.

On page 17 we answer this question through the following sentence: At least 50% of the Cr-Xbp1-downstream genes expressed in the *Ciona* notochord have been reported to participate in the UPR in other model organisms.

Could differential expression of UPR genes be a result of the way the experiment was done and the use of over expression constructs? Are these genes up or down regulated by the Bra>Xbp1dbd::GFP or Bra>Xbp1::VP16::GFP. I am aware that the control has GFP, but what is the size of the coding regions of the other constructs, how much was electroporated, I could not find the ug of constructs used for these experiments. I would like to know more about how the experiment was done to rule out the possibility that the UPR genes were differentially expressed as a result of electroporation of these constructs.

We have added the amount of constructs used for each experiment in the Methods section and figure legends. The coding region of GFP is ~720 bp (240 aa), the coding region of Xbp1DBD::GFP is approximately 1284 bp, for a total of ~428 amino acid residues. The coding region of Xbp1::VP16::GFP is ~1442 bp. We have used GFP as a control also in another published microarray screen, for Ciona Tbx2/3 (Jose’-Edwards et al., 2013). There is limited overlap between the targets of Tbx2/3 and those of Xbp1, and this overlap mostly includes Ciona/ascidian-specific genes that do not appear to be involved in UPR. This suggests that the results of these microarray screens are specific to each transcription factor and that the activation of UPR genes is not due to the electroporation procedure, but is rather due to the specific effect of Xbp1 on its transcriptional targets.

The text should clearly point out that their observations regarding the co-option of genes involved in the unfolded protein response in notochord development relates to Ciona only. In the manuscript the only data I see on genes downstream of Xbp1 studied in *Xenopus* is Shh. In the discussion they mention Bmp being known to be perturbed by Xbp1 knock down in *Xenopus* and that similar results were seen in Ciona.

The co-option of UPR genes in notochord development is suggested by the work of Tanegashima et al. (2009). In the Introduction (page 2) we have clarified that “Through the identification of Xbp1-downstream notochord genes in *Ciona*, we found evidence of the co-option of genes involved in the unfolded protein response to the notochord developmental program.”

It would be helpful to know if the genes identified as downstream of Xbp1 in the Ciona are also expressed in the *Xenopus* notochord. Could they have a figure showing what fraction of the 109 genes identified are also found in the *Xenopus* notochord?

To answer this question, we have added a supplemental table (Table S3) that contains the information currently available on *Xenopus* orthologs of *Ciona* Xbp1-downstream notochord genes.

Xbp1 knock out mice have been studied. In the publications about these mice are there any notochord defects mentioned? If in the knockout mice there are notochord defects this should be stated. If there are notochord defects this provides evidence in another vertebrate of the role of Xbp1 in notochord development.The authors state that Xbp1 in the mammalian notochord has not been studied. There is ample single cell RNA-seq datasets from embryos in mammalian systems in which the authors could look for notochord expression of Xbp1. It would help the manuscript if this data was utilized to identify if expression of this gene is conserved in other vertebrates.

We appreciate this helpful suggestion, and indeed we had been trying to find some information on the role of Xbp1 in the mouse notochord. We had contacted a few of the scientists involved in this published research to address this important point, but none of those who responded had a specific answer to our questions. Nevertheless, we have expanded the references on the mouse Xbp1 KO experiments and we now discuss the mouse single-cell RNA-Seq results and other data that tentatively suggest that Xbp1 might be involved in notochord formation in mouse embryos as well.

Since Xbp1 has already been studied in the *Xenopus* notochord, it would be helpful to state what has been learned from this study that was not appreciated in the previous analysis of Xbp1 in *Xenopus* to clearly show the knowledge added by this study.

In the Discussion section we clarify as follows: “the results of this study have uncovered a regulatory connection between Brachyury and Xbp1 that is maintained in *Xenopus* and is required for the proper development of the notochord in this vertebrate. The information gathered using *Ciona* on the genes influenced by Xbp1 will guide future studies on the notochord genes controlled by *Xenopus* Xbp1.”

There are several statements regarding data followed by data not shown. The data should be shown.

To address this point we have added two supplemental figures to this revised manuscript. Figure S2 shows microphotographs of ten additional novel notochord genes identified through this work (previously listed as ‘data not shown’ in Table S1). Figure S4 shows the expression of eight Xbp1 target genes with no published data, which we have found to be expressed in tissues other than the notochord (from Table S2). In addition, we have reorganized the figure showing Ciona/ascidian-specific notochord genes (now called Figure S3) and we have added a pie graph that summarizes the results of GO and InterPro searches. New or additional insets have been added to existing figures to provide additional information.

All three mentions of ‘data not shown’ that were present in the original text have been removed by filling in the gaps in information to the best of our abilities.

We have removed the expression patterns of genes for which we did not have convincing images or for which we were not able to repeat the in situ hybridizations, and therefore these genes are now listed as ‘not analyzed’ in Table S2 and in the text. This explains the change in the number of notochord genes in Table S1 and throughout the text, from 80 to 71.